# Continuous-time Particle Filtering for Latent Stochastic Differential Equations

## Abstract

Particle filtering is a standard Monte-Carlo approach for a wide range of sequential inference tasks. The key component of a particle filter is a set of particles with importance weights that serve as a proxy of the true posterior distribution of some stochastic process. In this work, we propose the adoption of continuous-time particle filtering for different types of inference tasks in neural latent stochastic differential equations. We demonstrate how such particle filters can be used as a generic plug-in replacement for inference techniques relying on a learned variational posterior. Our experiments with different model families based on neural latent stochastic differential equations demonstrate superior performance of continuous-time particle filtering in inference tasks like likelihood estimation and sequential prediction, both in synthetic and real-world scenarios.

## 1 Introduction

Over the last years neural architectures based on latent stochastic differential equations (latent SDEs; Tzen & Raginsky (2019); Li et al. (2020)) have emerged as expressive models of continuous-time dynamics with instantaneous noise. As a temporal backbone in continuously-indexed normalizing flows (Deng et al., 2021) they have also proven to be a powerful latent representation of non-Markovian dynamics. Further extensions to high-dimensional time-series (Hasan et al., 2020) and infinitely-deep Bayesian networks (Xu et al., 2022) have demonstrated applications in biology and computer vision.

Many important inference tasks in the latent SDE framework, including likelihood estimation and sequential prediction, are essentially filtering problems, as they require estimating the posterior distribution of the latent state from discrete and noisy observations, which can be viewed as partial realizations of an intrinsically continuous observation process. Since the posterior distribution in latent SDEs cannot be inferred exactly, the predominant approach in the existing literature is a variational approximation. Examples include Li et al. (2020), who introduce a learnable variational posterior SDE and derive an evidence lower bound (ELBO) (Kingma & Welling, 2013) of the observational log-likelihood. While intriguing at an architectural level, there is no guarantee how well this variational posterior process can capture the true posterior of the latent dynamics, which can lead to inaccurate inference. In another work, Deng et al. (2021) employ sequential importance weighting with multiple samples of the latent state, which can improve posterior estimation. However, the importance weights in this approach are prone to concentration on just a few samples as more observations are added over time, effectively reducing sample diversity.

A popular framework for filtering problems is particle filtering Gordon et al. (1993), which uses a set of particles (weighted samples) as a proxy of the true posterior. In contrast to a variational posterior, the posterior distribution represented by the particles converges asymptotically to the true posterior as we increase the number of samples. Furthermore, bootstrap resampling can be applied to the particles to improve their diversity and efficiency. In a discrete setting, Maddison et al. (2017) demonstrate the advantages of particle filtering over variational approximations in sequential inference tasks. For stochastic differential equations with simple forms, particle filtering has also been extended to the continuous-time domain Sottinen & Särkkä (2008). In this paper, we build on these ideas and develop a continuous-time inference framework that enables the use of particle filtering in *neural* architectures with *general* latent SDEs. In this context, continuous-time particle filtering serves as a drop-in replacement for inference based on sequential importance weighting(Le et al., 2017).

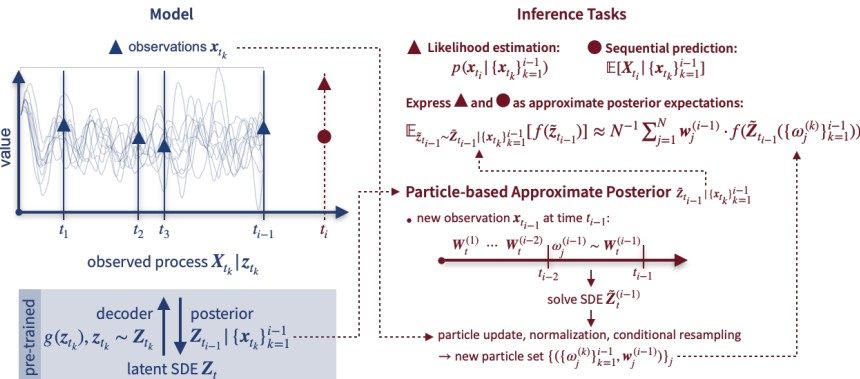

Figure 1: **Overview.** We consider inference tasks like likelihood estimation (▲) and sequential prediction (●) in a pre-trained model with latent SDE $\boldsymbol{Z}_t$ and observed process $\boldsymbol{X}_t$ connected by a neural decoder $g(\cdot)$. We are interested in performing inference conditioned on irregularly-spaced observations $\{\boldsymbol{x}_{t_k}\}_k$ at real-valued time points $\{t_k\}_k$. To this end, we express (▲) and (●) as posterior expectations and approximate the posterior with a continuous-time particle filter, whereby each time interval $[t_{k-1}, t_k]$ is represented by a separate posterior SDE $\tilde{\boldsymbol{Z}}_t^{(k)}$ with associated Wiener process $\boldsymbol{W}_t^{(k)}$. Given a new observation $\boldsymbol{x}_{t_{i-1}}$ at time $t_{i-1}$, we solve the corresponding posterior SDE $\tilde{\boldsymbol{Z}}_t^{(i-1)}$ and use the resulting latent points and importance weights – as well as the observations – to update our particle set. The new particles represent a revised posterior and, consequently, revised likelihood and prediction estimates.

First, we provide an introduction to (discrete-time) particle filtering and latent SDEs (Section 2). Then, we introduce a rigorous mathematical framework that defines the structure of particles, weights, and updates in the context of latent SDEs (Section 3.1). Finally, we demonstrate how important inference tasks in latent SDEs, such as likelihood estimation and sequential prediction, can be cast in the form of expectations over posterior distributions that are amenable to particle filtering (Section 3.2). The benefits of this approach are two-fold: (1) during likelihood estimation, the resampling step of the particle filter drops samples with smaller weights and keeps samples with larger weights, leading to *higher sample efficiency* than sequential importance weighting; (2) during sequential prediction, the particle filter converges in expectation to the true posterior, leading to *higher prediction accuracy* than a potentially restrictive learned approximation. A high-level overview of our framework's components and their interactions is shown in Figure 1. Our experiments (Section 4) validate continuous-time particle filtering for latent SDEs on stochastic processes with a broad range of properties, including geometric Brownian motion and SDEs with linear, multi-dimensional, coupled, and non-Markovian dynamics.

**Contributions.** In summary, we make the following contributions: (1) we propose the adoption of continuous-time particle filters during inference in latent SDE architectures; (2) we present a rigorous mathematical framework interpreting traditional particle filtering in the continuous-time domain based on the change-of-measure theorem for stochastic processes; (3) we demonstrate the use of the proposed continuous-time particle filtering framework as a plug-in replacement for importance-weighted variational inference in latent SDEs; and (4) we evaluate the resulting estimator on likelihood estimation and sequential prediction tasks, and demonstrate superior sample-efficiency and accuracy on a broad set of stochastic processes and real-world datasets.

## 2 Preliminaries

### 2.1 Particle Filtering

Particle filtering is the name given to a set of algorithms that solve the filtering problem, i.e., that provide a probabilistic estimate of a dynamic system's latent state given noisy observations ($\hat{=}$ posterior distribution). Instead of assuming a parametric form to describe the dynamic system's posterior distribution, particle filters use a set of weighted samples (particles) as a proxy, making it one of the most flexible approaches to the filtering problem. The posterior distribution represented by the particles is updated each time we make a new observation by executing the following three steps: (1) simulating new particles; (2) updating the weights of the particles; and (3) applying bootstrap resampling to the particles.

As an introductory example, consider a discrete-time system that is governed by latent dynamics $Z_i$ with transition probabilities $\pi_{Z_{i+1}|Z_i}(z_{i+1}|z_i)$ and noisy observations $X_i$ of $Z_i$ with probabilities $p_{X_i|Z_i}(x_i|z_i)$. To draw samples from $Z_i$, particle filtering relies on an importance sampling distribution $\gamma_{Z_{i+1}|Z_i}(z_{i+1}|z_i)$, which we assume here to be identical for all steps $i$. Given a set of $N$ particles $\{z_{1:i}^{(j)}\}_{j=1}^N$ with their respective weights $\{w_i^{(j)}\}_{j=1}^N$ and a new observation $x_{i+1}$, samples of $z_{i+1}^{(j)}$ are simulated from $\gamma_{Z_{i+1}|Z_i}$ for each $j$ and concatenated with $z_{1:i}^{(j)}$ to obtain the new particle $z_{1:i+1}^{(j)}$. The weights $\{w_i^{(j)}\}_{j=1}^N$ are then updated such that $w_{i+1}^{(j)} \propto w_i^{(j)} \cdot \frac{p_{X_i|Z_i}(x_i|z_i)\pi_{Z_{i+1}|Z_i}(z_{i+1}^{(j)}|z_i^{(j)})}{\gamma_{Z_{i+1}|Z_i}(z_{i+1}^{(j)}|z_i^{(j)})}$ and $\sum_{j=1}^N w_{i+1}^{(j)} = 1$. Finally, bootstrap resampling can be applied to the particles and the weights are reset to $\frac{1}{N}$. To obtain a Monte-Carlo estimate of an expectation over the posterior distribution, we use the set of particles as importance weighted samples of $Z_{1:i+1}$ and take a weighted average. As the number of weighted samples increases, the distribution represented by the particles will asymptotically converge to the true posterior distribution.

In the remainder of this paper, our focus is on a different type of dynamic system in which the latent dynamics are continuous in time and governed by a pre-trained neural SDE that is connected to the observation space through a neural decoder; see Section 2.2 below. Inference in such a system is based on discrete observations at irregular points in time. However, particle filtering algorithms for different dynamic systems all share the essence of using a set of weighted samples as a proxy of the posterior distribution.

## 2.2 Latent SDEs for Time-Series Modeling

Since our continuous-time particle filtering approach is an inference algorithm designed for models in the latent SDE family (Li et al., 2020; Deng et al., 2021), we will first provide an abstract description of the main ideas behind these models using a unified notation. Let $\{(t_i, \boldsymbol{x}_{t_i})\}_{i=1}^n$ denote a sequence of observations on the time grid $t_1 < t_2 < \cdots < t_n$, with $t_i \in (0, T)$, and $\boldsymbol{x}_{t_i}$ an $m$-dimensional observation at time point $t_i$. Furthermore, let $(\Omega, \mathcal{F}_t, P)$ be a filtered probability space on which $\boldsymbol{W}_t$ is a $d$-dimensional Wiener process. A latent stochastic differential equation modeling this observation sequence consists of two stochastic processes: a $d$-dimensional latent process $\boldsymbol{Z}_t$ defined by the stochastic differential equation

$$\mathrm{d}\boldsymbol{Z}_t = \mu_\theta(\boldsymbol{Z}_t, t)\,\mathrm{d}t + \sigma_\theta(\boldsymbol{Z}_t, t)\,\mathrm{d}\boldsymbol{W}_t, \tag{1}$$

and an $m$-dimensional observable process $\boldsymbol{X}_t = g_\theta(\boldsymbol{Z}_t)$ obtained by decoding the latent process with a decoder $g_\theta$, where $\theta$ denotes all parameters of the model. We use Itô integral as the definition for all the stochastic integrals in this work. It is worth noting that each stochastic differential equation can be viewed as a mapping of Wiener process paths to SDE paths, with the Wiener process being the actual source of stochasticity. To sample observation sequences $\{\boldsymbol{x}_{t_i}\}_{i=1}^n$ from the model, we first obtain samples $\{\boldsymbol{z}_{t_i}\}_{i=1}^n$ from the joint distribution of $\{\boldsymbol{Z}_{t_i}\}_{i=1}^n$ induced by the process $\boldsymbol{Z}_t$ on the given time grid by solving Eq.(1). The sample sequence $\{\boldsymbol{z}_{t_i}\}_{i=1}^n$ is then decoded into a joint observational distribution over $\{\boldsymbol{X}_{t_i}\}_{i=1}^n$ conditioned on the values $\{\boldsymbol{z}_{t_i}\}_{i=1}^n$. In some models (e.g., Li et al. (2020)), the sample sequence $\{\boldsymbol{z}_{t_i}\}_{i=1}^n$ can be decoded independently at each time step $t_i$,

$$p_{\boldsymbol{X}_{t_1},\ldots,\boldsymbol{X}_{t_n}|\boldsymbol{Z}_{t_1},\ldots,\boldsymbol{Z}_{t_n}}(\boldsymbol{x}_{t_1},\ldots,\boldsymbol{x}_{t_n}|\boldsymbol{z}_{t_1},\ldots,\boldsymbol{z}_{t_n}) = \prod_{i=1}^n p_{\boldsymbol{X}_{t_i}|\boldsymbol{Z}_{t_i}}(\boldsymbol{x}_{t_i}|\boldsymbol{z}_{t_i}), \tag{2}$$

while other models (e.g., Deng et al. (2021)) assume more complex dependency structures between the $\boldsymbol{Z}_{t_i}$'s and $\boldsymbol{X}_{t_i}$'s.

### 2.2.1 Importance Weighting for Latent SDEs

Similar to other latent variable models (Kingma & Welling, 2013), computing the marginal observation likelihood in latent SDEs is intractable as it requires the integral with respect to the distribution of the Wiener process $\boldsymbol{W}_t$, which is the ultimate source of stochasticity in the latent dynamics. Due to the intractability of this integral, training and inference traditionally rely on a learned variational posterior process and importance sampling. The learned variational posterior process induces a distribution that assigns weights to the trajectories of the Wiener process in a way that differs from the original distribution. Driven by

Wiener process trajectories sampled from this induced distribution, the latent dynamics $\boldsymbol{Z}_t$ have a higher likelihood of reconstructing the observations $\{\boldsymbol{x}_{t_i}\}_{i=1}^n$. Girsanov's Theorem Oksendal (2013) establishes the theoretical basis of computing the ratio between the weights of the two distributions (importance weight). In a broader sense, importance sampling is an important tool in Monte-Carlo methods to estimate expectations of functions of latent states. Likewise, it is also an essential component of particle filtering. In the remainder of this section we derive the importance weight induced by a variational posterior process and simultaneously lay the foundation for the particle filtering approach discussed in Section 3.

Given the observations $\{(t_i, \boldsymbol{x}_{t_i})\}_{i=1}^n$, the posterior process $\tilde{\boldsymbol{Z}}_t$ in a latent SDE model is characterized by a stochastic differential equation with an observation-dependent drift term $\mu_\phi$ and a shared variance term $\sigma_\theta$,

$$\mathrm{d}\tilde{\boldsymbol{Z}}_t = \mu_\phi(\tilde{\boldsymbol{Z}}_t, t) \ \mathrm{d}t + \sigma_\theta(\tilde{\boldsymbol{Z}}_t, t) \ \mathrm{d}\boldsymbol{W}_t, \tag{3}$$

such that Eq.(3) satisfies Novikov's condition

$$\mathbb{E}\left[\exp(\int_0^T \frac{1}{2} \left|u(\tilde{\boldsymbol{Z}}_t, t)\right|^2 \ \mathrm{d}t)\right] < \infty, \tag{4}$$

with $\sigma_\theta(z,t)u(z,t) = \mu_\phi(z,t) - \mu_\theta(z,t)$. The parameters of the drift function $\phi$ are produced by an encoder taking the observations $\{(t_i, \boldsymbol{x}_{t_i})\}_{i=1}^n$ as inputs, so the posterior process can be parameterized by the observations. By using observations to parameterize $\phi$ and optimizing the evidence lower bound (ELBO) of the log-likelihood, the variational posterior process is explicitly encouraged to encode the observations into latent dynamics for reconstructions with high likelihood. It can therefore be used for a variety of downstream inference tasks, including likelihood estimation and sequential prediction. The parameters of the variance function $\theta$ are the same as in the prior latent process (Eq.(1)) to allow computation of the importance weight between the prior and posterior process as follows: by Girsanov's Theorem (Oksendal, 2013, Theorem 8.6.4), we can reweight the trajectories of $\boldsymbol{W}_t$ by replacing the distribution $P$ of the probability space $(\Omega, \mathcal{F}_t, P)$ with a different distribution $Q$ such that, with $\boldsymbol{W}_t$ defined on $(\Omega, \mathcal{F}_t, Q)$, $\tilde{\boldsymbol{W}}_t = \int_0^t u(\tilde{\boldsymbol{Z}}_t, t) \ \mathrm{d}t + \boldsymbol{W}_t$ is another Wiener process. Given a sample $\omega \in \Omega$, the importance weight (Radon-Nikodym derivative) $\boldsymbol{M}_t(\omega)$ between the distributions $Q$ and $P$ can now be written as

$$\boldsymbol{M}_t(\omega) = \exp(-\int_0^t \frac{|u(\tilde{\boldsymbol{Z}}_s(\omega), s)|^2}{2} \ \mathrm{d}s - \int_0^t u(\tilde{\boldsymbol{Z}}_s(\omega), s)^T \ \mathrm{d}\boldsymbol{W}_s(\omega)). \tag{5}$$

Using the definition of $\tilde{\boldsymbol{W}}_t$, Eq.(3) can be rewritten as

$$\mathrm{d}\tilde{\boldsymbol{Z}}_t = \mu_\phi(\tilde{\boldsymbol{Z}}_t, t) \ \mathrm{d}t + \sigma_\theta(\tilde{\boldsymbol{Z}}_t, t) \ \mathrm{d}\boldsymbol{W}_t = \mu_\theta(\tilde{\boldsymbol{Z}}_t, t) \ \mathrm{d}t + \sigma_\theta(\tilde{\boldsymbol{Z}}_t, t) \ \mathrm{d}\tilde{\boldsymbol{W}}_t. \tag{6}$$

Since $\tilde{\boldsymbol{W}}_t$ defined on $(\Omega, \mathcal{F}_t, Q)$ is also a Wiener process, the distribution of $\{\tilde{\boldsymbol{Z}}_{t_i}\}_{i=1}^n$ under $Q$ is the same as the distribution of $\{\boldsymbol{Z}_{t_i}\}_{i=1}^n$ under $P$.

Sampling directly from the unknown distribution $Q$ is challenging. However, to obtain a Monte-Carlo estimate with respect to $Q$, we can first sample $\boldsymbol{W}_t(\omega)$ using the distribution $P$ and then apply the importance weight $\boldsymbol{M}_t$ defined by Eq.(5), leading to

$$\mathbb{E}_P[f(\{\boldsymbol{Z}_{t_i}\}_{i=1}^n)] = \mathbb{E}_Q[f(\{\tilde{\boldsymbol{Z}}_{t_i}\}_{i=1}^n)] = \mathbb{E}_P[f(\{\tilde{\boldsymbol{Z}}_{t_i}\}_{i=1}^n)\boldsymbol{M}_t], \tag{7}$$

where $f(\{\boldsymbol{Z}_{t_i}\}_{i=1}^n)$ is a function whose expected value we are interested in. The posterior process $\tilde{\boldsymbol{Z}}_t$ and the importance weighting process $\boldsymbol{M}_t$ can be concatenated and together form the solution to an augmented version of the SDE characterizing the posterior process (Eq.(3)).

## 3 Continuous-Time Particle Filtering for Latent SDEs

With the preliminaries on latent SDEs and importance weighting introduced, we will now present our continuous-time particle filtering approach (Section 3.1), along with two applications to inference in latent SDEs (Section 3.2).

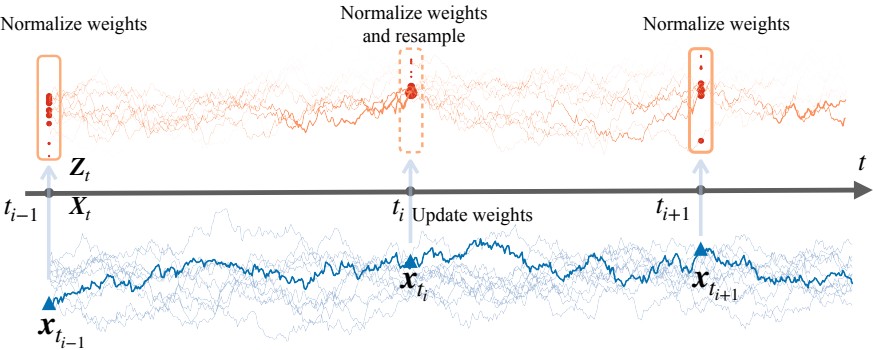

Figure 2: **Continuous-Time Particle Filtering.** Given discrete observations (blue triangles) of a sample trajectory from the observed process $\boldsymbol{X}_t$, the importance weights of latent trajectories (orange trajectories), represented by line thickness, are first updated by observation likelihood (blue arrows) and then normalized (orange boxes). If the normalized weights (orange circles) are concentrated on a small subset of the particles, the particles will be resampled (dashed orange box), with the normalized weights set to be uniform in value. The normalized weights are used as initial weights of the latent trajectories in the following time interval.

### 3.1 Particles and Weights for Continuous-time Latent SDEs

The core intuition behind particle filtering is to use a set of particles (i.e., weighted samples from a distribution) as a proxy of the posterior distribution of a stochastic process and the particles and their weights are updated sequentially given new observations. Three important questions need to be answered when extending particle filtering to the latent SDE framework: (1) what are the particles in a continuous-time particle filter; (2) what are the importance distribution and weights of particles from the distribution; and (3) how should the particle weights be updated. We begin by breaking the Wiener process trajectories into pieces and take these pieces of sampled trajectories as particles. This piece-wise construction also allows us to apply importance weighting individually to each interval of the latent dynamics and new observations. Together, these design choices form the foundation of continuous-time particle filtering for latent SDEs. We conclude this section by describing the adoption of standard particle filtering approaches, including weight update, normalization, and bootstrap resampling in the context of the proposed framework.

**Continuous-time Particles.** In the latent process of latent SDE models, the expectation in Eq.(7) is taken over the distribution of a Wiener process. It is thus natural to use sample trajectories of a Wiener process as particles in the particle filter. Even though Wiener processes are usually defined on probability spaces with continuous filtration, the resampling step of a particle filter is a discrete event. As a consequence, we need to give the Wiener process and the continuous-time stochastic processes induced by latent SDEs sequential structure to define particles in a continuous-time latent SDE setting. To this end, we leverage the piece-wise construction of Wiener processes proposed in Deng et al. (2021): given the time grid $0 = t_0 < t_1 < \cdots < t_n = T$, where each $t_i$ is the observation time point of $\boldsymbol{x}_{t_i}$, we can obtain a sample trajectory of a Wiener process $\boldsymbol{W}_t$ of length $T$ by sampling trajectories from $n$ independent Wiener processes $\boldsymbol{W}_t^{(i)}$, each with length $t_i - t_{i-1}$, defined on the probability space $(\Omega^{(i)}, \mathcal{F}_t^{(i)}, P^{(i)})$ and add them together,

$$\boldsymbol{W}_t(\omega^{(1)}, \omega^{(2)}, \ldots, \omega^{(n)}) = \sum_{\{i: t_i < t\}} \boldsymbol{W}_{t_i - t_{i-1}}^{(i)}(\omega^{(i)}) + \boldsymbol{W}_{t - t_{i^*}}^{(i^*)}(\omega^{(i^*)}), \tag{8}$$

where $i^* = \max\{i : t_i < t\} + 1$ and $\omega^{(i)} \in \Omega^{(i)}$. We can solve Eq.(1) in a similar piece-wise manner,

$$\boldsymbol{Z}_t = \sum_{\{i: t_i < t\}} \boldsymbol{Z}_{t_i} + \int_{t_{i^*}}^t \mu_\theta(\boldsymbol{Z}_s, s)\, \mathrm{d}s + \int_{t_{i^*}}^t \sigma_\theta(\boldsymbol{Z}_s, s)\, \mathrm{d}\boldsymbol{W}_{s - t_{i^*}}^{(i^*)}, \tag{9}$$

and rewrite $\mathbb{E}_P[f(\{\boldsymbol{Z}_{t_i}\}_{i=1}^n)]$ in Eq.( 7) as

$$\mathbb{E}_{P^{(1)} \times \cdots \times P^{(i)} \times \cdots \times P^{(n)}} \left[ f\left(\{\boldsymbol{Z}_{t_i}\}_{i=1}^n\right) \right] = \mathbb{E}_{P^{(1)}} \left[ \ldots \mathbb{E}_{P^{(i)}} \left[ \ldots \mathbb{E}_{P^{(n)}} \left[ f\left(\{\boldsymbol{Z}_{t_i}\}_{i=1}^n\right) \right] \ldots \right] \ldots \right]. \tag{10}$$

Similarly, the posterior process $\tilde{\boldsymbol{Z}}_t$ (Eq.(3)) can be obtained by defining $(\Omega^{(i)}, \mathcal{F}_t^{(i)}, Q^{(i)})$ and posterior process parameters $\phi^{(i)}$. We refer to the supplementary material for additional details on the piecewise construction

of the posterior process $\tilde{\boldsymbol{Z}}_t$. In summary, each particle in the continuous-time particle filter will be represented as a sequence of $\omega^{(i)}$'s, where each $\omega^{(i)}$ is a sample from $\Omega^{(i)}$.

**Importance Distribution and Weighting.** As it is usually difficult to directly sample from the target posterior distribution, particle filtering relies on an importance distribution, also called proposal distribution, for sampling. The importance distribution should be easy to sample from. Therefore, we use $(\Omega^{(i)}, \mathcal{F}_t^{(i)}, P^{(i)})$ as the importance distribution to sample $\omega^{(i)}$ in the interval $[t_{i-1}, t_i]$. This choice of importance distribution also admits a sequential structure in the sense that the distributions before $t_i$ will not be modified given future observations after $t_i$. Moreover, the evidence lower bound also encourages the variational posterior process to reconstruct observations with high likelihood from the Wiener process. We make the choice of $Q^{(i)}$ conditioned on samples from $\{\tilde{\boldsymbol{Z}}_{t_k}\}_{k<i}$ as the prior distribution for computing the weights of particles. It is worth noting that, despite being observation dependent, the $Q^{(i)}$'s induce the same finite-dimensional (prior and posterior) distributions of $\tilde{\boldsymbol{Z}}_t$ as the finite-dimensional distribution of $\boldsymbol{Z}_t$ induced by the $P^{(i)}$'s. As a result, we can rewrite Eq.(7) using the piece-wise approach as follows:

$$\mathbb{E}_{P^{(1)}\times\cdots\times P^{(i)}\times\cdots\times P^{(n)}}\left[f\left(\{\boldsymbol{Z}_{t_i}\}_{i=1}^n\right)\right] = \mathbb{E}_{P^{(1)}}\left[\ldots\mathbb{E}_{P^{(i)}}\left[\ldots\mathbb{E}_{P^{(n)}}\left[f\left(\{\boldsymbol{Z}_{t_i}\}_{i=1}^n\right)\right]\ldots\right]\ldots\right]$$

$$= \mathbb{E}_{Q^{(1)}|\{\tilde{\boldsymbol{z}}_{t_0}\}}\left[\ldots\mathbb{E}_{Q^{(i)}|\{\tilde{\boldsymbol{z}}_{t_k}\}_{k=1}^{i-1}}\left[\ldots\ \mathbb{E}_{Q^{(n)}|\{\tilde{\boldsymbol{z}}_{t_k}\}_{k=1}^{n-1}}\left[f(\{\tilde{\boldsymbol{Z}}_{t_i}\}_{i=1}^n)\right]\ldots\right]\ldots\right], \quad (11)$$

where $P^{(1)}\times\cdots\times P^{(i)}\times\cdots\times P^{(n)}$ is the product distribution of all $P^{(i)}$'s from which we sample the $\boldsymbol{W}_t^{(i)}$'s. To compute the (unnormalized) weights of particles from the importance distribution $(\Omega^{(i)}, \mathcal{F}_t^{(i)}, P^{(i)})$, we also need to compute importance weights between each pair of $Q^{(i)}|\{\tilde{\boldsymbol{Z}}_{t_k}\}_{k=1}^{i-1}$ and $P^{(i)}$. Using Eq.(7), the $i$-th nested expectation in Eq.(11) can be rewritten as

$$\mathbb{E}_{Q^{(i)}|\{\tilde{\boldsymbol{z}}_{t_k}\}_{k=1}^{i-1}}\left[\ldots\mathbb{E}_{Q^{(n)}|\{\tilde{\boldsymbol{z}}_{t_k}\}_{k=1}^{n-1}}\left[f(\{\tilde{\boldsymbol{Z}}_{t_i}\}_{i=1}^n)\right]\ldots\right] = \mathbb{E}_{P^{(i)}}\left[\ldots\mathbb{E}_{P^{(n)}}\left[f(\{\tilde{\boldsymbol{Z}}_{t_i}\}_{i=1}^n)\boldsymbol{M}^{(n)}\right]\ldots\boldsymbol{M}^{(i)}\right]. \quad (12)$$

Here we use $\boldsymbol{M}^{(i)}$ as a shorthand for $\boldsymbol{M}^{(i)}(\omega^{(i)}|\{\omega^{(k)}\}_{k=1}^{i-1})$, i.e., the importance weight term of sample $\omega^{(i)}$ from $(\Omega^{(i)}, \mathcal{F}_t^{(i)}, P^{(i)})$ conditioned on $\{\omega^{(k)}\}_{k=1}^{i-1}$.

**Particle Updates.** With the proposal distribution and importance weighting specified, we can now present how samples are obtained and weights are updated in continuous-time particle filtering given a sequence of observations $\boldsymbol{x}_{t_i}$. Let $N$ be the number of particles in a particle filter and $\{(\{\omega_j^{(k)}\}_{k=1}^i, \boldsymbol{w}_j^{(i)})\}_{j=1}^N$ denote the set of particles after the $i$-th observation $\boldsymbol{x}_{t_i}$ and update, where $\omega_j^{(k)}$ is a sample from $(\Omega^{(k)}, \mathcal{F}_t^{(k)}, P^{(k)})$ for each $j$ and $\boldsymbol{w}_j^{(i)}$ is the weight of the $j$-th particle up to time $t_i$. Following the standard formulation of particle filtering, we initialize the set of particles as $\{(\{\}, \boldsymbol{w}_j^{(0)})\}_{j=1}^N$, i.e., each particle has no sample, denoted as $\{\}$, and an initial weight of $\boldsymbol{w}_j^{(0)} := \frac{1}{N}$. The set of particles and their corresponding weights are updated after each new observation. Specifically, given the set of particles $\{(\{\omega_j^{(k)}\}_{k=1}^{i-1}, \boldsymbol{w}_j^{(i-1)})\}_{j=1}^N$ at time point $t_{i-1}$ and the $i$-th observation $\boldsymbol{x}_{t_i}$, the particles are updated as follows:

$$\{\omega_j^{(k)}\}_{k=1}^i \leftarrow CONCAT\left(\{\omega_j^{(k)}\}_{k=1}^{i-1}, \{\omega_j^{(i)}\}\right),$$

$$\tilde{\boldsymbol{w}}_j^{(i)} \leftarrow \boldsymbol{w}_j^{(i-1)} p(\boldsymbol{x}_{t_i}|\{\boldsymbol{x}_{t_k}\}_{k=1}^{i-1}, \{\omega_j^{(k)}\}_{k=1}^i)\boldsymbol{M}_j^{(i)}, \quad (13)$$

$$\boldsymbol{w}_j^{(i)} \leftarrow \tilde{\boldsymbol{w}}_j^{(i)} / \sum_{j=1}^N \tilde{\boldsymbol{w}}_j^{(i)}$$

where $\boldsymbol{M}_j^{(i)}$ is the importance weight of sample $\omega_j^{(i)}$ in the interval $t_i - t_{i-1}$ and $p(\boldsymbol{x}_{t_i}|\{\boldsymbol{x}_{t_k}\}_{k=1}^{i-1}, \{\omega_j^{(k)}\}_{k=1}^i)$ is the observation likelihood used to update the weights of particles conditioned on previous observations and latent samples. The likelihood term can usually be written as $p(\boldsymbol{x}_{t_i}|\{\boldsymbol{x}_{t_k}\}_{k=1}^{i-1}, \{\boldsymbol{z}_{t_k,j}\}_{k=1}^i)$, with $\boldsymbol{z}_{t_k,j}$ defined by $\omega_j^{(k)}$ through the latent SDEs.

When certain criteria are met (Doucet et al., 2009), we resample the particles from the categorical distribution defined by the particle weights and all the weights $\boldsymbol{w}_j^{(i)}$ are reset to $\frac{1}{N}$. In our experiments, we chose to trigger the resampling when the effective sample size of the particles drop below half of $N$ after each new observation. The process of sampling and updating particle weights is visualized in Figure 2 and a detailed presentation of our particle filtering algorithm in pseudocode is included in the supplementary material.

### 3.2 Inference with Continuous-time Particle Filtering

Many inference tasks relying on the posterior distribution can be expressed as expectations w.r.t. certain functions over the posterior distribution of $Q$ conditioned on observations $\{\boldsymbol{x}_{t_i}\}_{i=1}^n$, i.e., $\mathbb{E}_{Q|\{\boldsymbol{x}_{t_i}\}_{i=1}^n}[f(\{\tilde{\boldsymbol{Z}}_{t_i}\}_{i=1}^n)]$. As the set of particles with their weights is a proxy of the posterior distribution $Q|\{\boldsymbol{x}_{t_i}\}_{i=1}^n$, the weighted average of the function over the particles is a Monte-Carlo integration and an estimator of the expectation. We present two applications of continuous-time particle filtering based on this principle: likelihood estimation and sequential prediction.

**Likelihood Estimation.** In latent variable models, it is a common practice to approximate the observation log-likelihood using an IWAE bound with multiple latent samples (Burda et al., 2015). When applied to latent SDE models (Li et al., 2020; Deng et al., 2021), the IWAE bound can be viewed as a specific instance of sequential importance sampling. One concern with sequential importance sampling is decreasing sampling efficiency as time increases, e.g., as a result of importance weights becoming skewed over time, with most weights concentrated on a few samples. The resampling step of particle filters can remove samples with smaller importance weights while preserving the ones with larger importance weights. Given a sequence of observations $\{\boldsymbol{x}_{t_i}\}_{i=1}^n$, we take the integrand function $f$ of $\{\tilde{\boldsymbol{Z}}_{t_k}\}_{k=1}^{i-1}$ at time $t_{i-1}$ to be $p(\boldsymbol{x}_{t_i}|\{\boldsymbol{x}_{t_k}\}_{k=1}^{i-1}, \{\tilde{\boldsymbol{Z}}_{t_k}\}_{k=1}^{i-1})$, leading to

$$p(\boldsymbol{x}_{t_i}|\{\boldsymbol{x}_{t_k}\}_{k=1}^{i-1}) = \mathbb{E}_{Q|\{\boldsymbol{x}_{t_i}\}_{i=1}^{k-1}}[p(\boldsymbol{x}_{t_i}|\{\boldsymbol{x}_{t_k}\}_{k=1}^{i-1}, \{\tilde{\boldsymbol{Z}}_{t_k}\}_{k=1}^{i-1})]. \tag{14}$$

The integrand function can be estimated as follows:

$$p(\boldsymbol{x}_{t_i}|\{\boldsymbol{x}_{t_k}\}_{k=1}^{i-1}, \{\tilde{\boldsymbol{Z}}_{t_k}\}_{k=1}^{i-1}) = \mathbb{E}_{P^{(i)}}[p(\boldsymbol{x}_{t_i}|\{\boldsymbol{x}_{t_k}\}_{k=1}^{i-1}, \{\tilde{\boldsymbol{Z}}_{t_k}\}_{k=1}^{i})\boldsymbol{M}_{t_i-t_{i-1}}^{(i)}], \tag{15}$$

where each $\tilde{\boldsymbol{Z}}_{t_k}$ is a function of $\{\omega^{(l)}\}_{l=1}^k$. The samples used for estimation in Eq.(15) can in turn be reused to update the particles at time $t_i$.

**Sequential Prediction.** Many sequential latent variable models (Rubanova et al., 2019; Li et al., 2020; Deng et al., 2020) rely solely on the proposal distribution for forecasting and completely discard the prior distribution, which also defines the true posterior distribution. Using particle filtering for forecasting not only reintroduces the prior distribution to the inference task but also makes use of a proxy version of the true posterior instead of an arbitrary proposal distribution learned from data. In particular, we are interested in the sequential prediction task of estimating the expectation of $\boldsymbol{X}_{t_{i+j}}$ for some $j \geqslant 1$ conditioned on the observations $\{\boldsymbol{x}_{t_k}\}_{k=1}^i$,

$$\mathbb{E}_{Q|\{\boldsymbol{x}_{t_k}\}_{k=1}^i}\left[\boldsymbol{X}_{t_{i+j}}\right] = \mathbb{E}_{Q|\{\boldsymbol{x}_{t_k}\}_{k=1}^i}\left[\mathbb{E}_{\boldsymbol{X}_{t_{i+j}}|\tilde{\boldsymbol{z}}_{t_i}, \{\boldsymbol{x}_{t_k}\}_{k=1}^i}\left[\boldsymbol{X}_{t_{i+j}}\right]\right], \tag{16}$$

where the expected value of $\boldsymbol{X}_{t_{i+j}}$ is conditioned on previous observations and the value of $\tilde{\boldsymbol{Z}}_{t_i}$ is the integrand of interest. The integrand can, in turn, be estimated by sampling $\tilde{\boldsymbol{Z}}_{t_{i+j}}$ from the prior and averaging the expectation of $\boldsymbol{X}_{t_{i+j}}$ conditioned on $\tilde{\boldsymbol{Z}}_{t_{i+j}}$ and $\{\boldsymbol{x}_{t_k}\}_{k=1}^i$.

## 4 Experiments

To demonstrate the effectiveness of the proposed approach, we compare continuous-time particle filtering-based inference in CLPF (Deng et al., 2021) and latent SDE (Li et al., 2020) models against alternative methods, including variational approximations and sequential importance sampling. We report results for two inference tasks, likelihood estimation and sequential prediction, on synthetic datasets simulated using common SDEs and real-world datasets.

### 4.1 Experiment Setting

Since we propose continuous-time particle filtering as an *inference* method for the latent SDE framework, we compare it against other inference approaches on the same pre-trained CLPF and latent SDE models

for each dataset; we refer to the supplementary material for additional details on model pre-training. In our likelihood estimation experiments, we compare the negative log-likelihood (NLL) estimates based on a standard variational approach (IWAE; (Burda et al., 2015)) and particle filtering. We note that IWAE can also be interpreted as a sequential importance sampling method in the likelihood estimation task. Because both IWAE estimation and particle filtering approximate the negative log-likelihood with upper bounds, lower results indicate better estimation. In sequential prediction tasks, we use the model to predict $\{\boldsymbol{x}_{t_i}\}_{i=j+1}^{j+k}$ given observations $\{(t_i, \boldsymbol{x}_{t_i})\}_{i=1}^{j}$ and target time stamps $\{t_i\}_{i=j+1}^{j+k}$, for $i \in \{2, \ldots, n-k\}$, where $k$ is the number of steps to predict ahead and $n$ is the length of the data sequence. We compare three sequential prediction approaches: particle filtering as introduced in Section 3.2, sequential importance sampling, which is similar to particle filtering but does not apply resampling, and a variational approximation, which is a common approach for prediction in latent variable models (Rubanova et al., 2019; Li et al., 2020; Deng et al., 2021) and uses the variational posterior process. In all experiments we use 125 latent samples for inference.

## 4.2 Synthetic Data Experiments

We evaluate all inference methods and models using asynchronous sequential samples simulated from four common continuous-time stochastic processes, i.e., the observation time stamps are on an irregular time grid. For each process, the observation time stamps for evaluation are sampled from homogeneous Poisson point processes with two different intensity values $\lambda$ (either $\{2, 20\}$ or $\{20, 40\}$), with larger values corresponding to denser observations in a time interval. The forecasting horizon in the sequential prediction task is set to one. We consider the following processes:[1]

**Geometric Brownian Motion (GBM).** Geometric Brownian motion is a stochastic process satisfying $\mathrm{d}\boldsymbol{X}_t = \mu \boldsymbol{X}_t \, \mathrm{d}t + \sigma \boldsymbol{X}_t \, \mathrm{d}\boldsymbol{W}_t$, i.e., the logarithm of geometric Brownian motion is Brownian motion.

**Linear SDE (LSDE).** In a linear SDE the drift term is a linear transformation and the variance term is a deterministic function of time $t$; it can be characterized as $\mathrm{d}\boldsymbol{X}_t = (a(t)\boldsymbol{X}_t + b(t)) \, \mathrm{d}t + \sigma(t) \, \mathrm{d}\boldsymbol{W}_t$.

**Continuous AR(4) Process (CAR).** The continuous autoregressive process of 4-th order can be viewed as the 1-dim. projection of the 4-dim. stochastic differential equation

$$\begin{aligned} \boldsymbol{X}_t &= [d, 0, 0, 0]\boldsymbol{Y}_t, \\ \mathrm{d}\boldsymbol{Y}_t &= A\boldsymbol{Y}_t \, \mathrm{d}t + e \, \mathrm{d}\boldsymbol{W}_t, \end{aligned} \text{ where } A = \begin{pmatrix} \mathbf{0} & \mathbf{I}_3 \\ a_1 & a_2 \quad a_3 \quad a_4 \end{pmatrix}.$$

**Stochastic Lorenz Curve (SLC).** The stochastic Lorenz curve is a three-dimensional continuous-time stochastic process characterized by the following system of equations:

$$\begin{aligned} \mathrm{d}\boldsymbol{X}_t &= \sigma(\boldsymbol{Y}_t - \boldsymbol{X}_t) \, \mathrm{d}t + \alpha_x \, \mathrm{d}\boldsymbol{W}_t, \\ \mathrm{d}\boldsymbol{Y}_t &= (\boldsymbol{X}_t(\rho - \boldsymbol{Z}_t) - \boldsymbol{Y}_t) \, \mathrm{d}t + \alpha_y \, \mathrm{d}\boldsymbol{W}_t, \\ \mathrm{d}\boldsymbol{Z}_t &= (\boldsymbol{X}_t\boldsymbol{Y}_t - \beta\boldsymbol{Z}_t) \, \mathrm{d}t + \alpha_z \, \mathrm{d}\boldsymbol{W}_t. \end{aligned}$$

**Quantitative Results.** Our likelihood estimation and sequential prediction results on synthetic data are shown in Table 1 and Table 2, respectively. We observe that the inference results based on particle filtering are better than the results obtained with other inference approaches in almost all settings (81% of test cases), regardless of model, task, and dataset. In particular, we find that particle filtering (with resampling) significantly outperforms IWAE (without resampling). Our results in the likelihood estimation task indicate that particle filtering algorithms generally perform better in challenging settings with $\lambda = 20$, including GBM and CAR, a non-Markov process. In the sequential prediction task we observe a similar trend, especially when the value of $\lambda$ is small and the observations sparse. In the experiments where inference based on particle filtering does not outperform the other approaches, its performance is still competitive. For large values of $\lambda$, previous observations can be close to the observation at the next time point because the change of a continuous process is small during a short interval, constraining the solution space. We hypothesize that this is also the reason why we see similar prediction accuracies of CLPF models with and without particle filtering with these settings, e.g., for GBM ($\lambda = 20$) and LSDE ($\lambda = [2, 20]$), as CLPF makes future predictions based on continuous trajectories extrapolated from given observations.

---

[1]Additional details on data generation, including parameter settings, can be found in the supplementary material.

Table 1: **Likelihood Estimation**. We compare the negative log-likelihoods (NLLs) of observations estimated using an IWAE approximation (IWAE) and our proposed particle filter (PF). We evaluate both techniques on four stochastic processes with two different observation intensity values $\lambda$. We report the mean and standard deviation over 5 runs. [GBM: geometric Brownian motion (ground truth NLLs: $[\lambda = 2, \lambda = 20] = [0.388, -0.788]$); LSDE: linear SDE; CAR: continuous auto-regressive process; SLC: stochastic Lorenz curve]

| Model (Inference) | GBM | | LSDE | | CAR | | SLC | |
|---|---|---|---|---|---|---|---|---|
| | $\lambda = 2$ | $\lambda = 20$ | $\lambda = 2$ | $\lambda = 20$ | $\lambda = 2$ | $\lambda = 20$ | $\lambda = 20$ | $\lambda = 40$ |
| CLPF (IWAE) | 0.444±0.001 | -0.698±0.000 | -0.831±0.001 | -1.939±0.000 | 1.321±0.002 | -0.077±0.000 | -2.620±0.001 | -3.962 ±0.001 |
| CLPF (PF) | **0.422±0.000** | **-0.756±0.000** | **-0.840±0.000** | **-1.985±0.000** | **1.213±0.000** | **-0.197±0.000** | **-2.647±0.001** | **-3.964±0.004** |
| Latent SDE (IWAE) | **1.242±0.001** | 1.777±0.002 | 0.082±0.001 | 0.218±0.002 | 3.593±0.002 | 3.610±0.005 | 7.738±0.007 | 8.257±0.001 |
| Latent SDE (PF) | 1.264±0.002 | **1.001±0.002** | **0.049±0.003** | **0.155±0.002** | **3.568±0.003** | **3.460±0.011** | **7.725±0.003** | **8.255±0.001** |

Table 2: **Sequential Prediction**. We report the average L2-distance between predictions and ground truth observations in a one-step-ahead sequential prediction setting. All predictions are based on the average of 125 latent samples. For each dataset, we report the mean value in the first row and the 25th and 75th percentiles of the prediction errors in the second row. [VA = variational approx.; SIS = sequential sampling sampling; PF = particle filtering]

| Model (Inference) | GBM | | LSDE | | CAR | | SLC | |
|---|---|---|---|---|---|---|---|---|
| | $\lambda = 2$ | $\lambda = 20$ | $\lambda = 2$ | $\lambda = 20$ | $\lambda = 2$ | $\lambda = 20$ | $\lambda = 20$ | $\lambda = 40$ |
| CLPF (VA) | 0.705 | **0.206** | **0.102** | **0.031** | 1.322 | 0.273 | 0.446 | **0.231** |
| | [0.077, 0.602] | **[0.022, 0.175]** | **[0.028, 0.043]** | **[0.009, 0.043]** | [0.095, 1.189] | [0.019, 0.208] | [0.079, 0.494] | **[0.045, 0.260]** |
| CLPF (SIS) | 0.811 | 0.296 | 0.109 | 0.040 | 1.556 | 0.538 | 0.507 | 0.316 |
| | [0.084, 0.680] | [0.028, 0.242] | [0.030, 0.147] | [0.011, 0.055] | [0.157, 1.444] | [0.047, 0.441] | [0.117, 0.585] | [0.079, 0.362] |
| CLPF (PF) | **0.693** | **0.206** | 0.103 | **0.031** | **0.753** | **0.119** | **0.422** | 0.236 |
| | **[0.078, 0.609]** | **[0.022, 0.176]** | [0.028, 0.139] | **[0.009, 0.043]** | **[0.069, 0.683]** | **[0.01, 0.092]** | **[0.077, 0.475]** | [0.046, 0.262] |
| Latent SDE (VA) | 1.836 | **1.066** | 0.302 | 0.177 | 63.750 | 57.212 | 14.356 | 14.210 |
| | [0.298, 1.600] | **[0.166, 0.879]** | [0.108, 0.420] | [0.065, 0.247] | [5.096, 55.093] | [2.581, 34.261] | [10.862, 13.773] | [10.691, 13.819] |
| Latent SDE (SIS) | 1.722 | 2.001 | 0.252 | 0.265 | 48.012 | 52.124 | 16.061 | 16.506 |
| | [0.262, 1.472] | [0.210, 1.443] | [0.088, 0.352] | [0.093, 0.373] | [2.317, 26.573] | [1.384, 25.213] | [9.992, 19.328] | [9.393, 18.943] |
| Latent SDE (PF) | **1.503** | 1.282 | **0.202** | **0.154** | **45.403** | **42.451** | **14.137** | **13.663** |
| | **[0.209, 1.261]** | [0.132, 0.884] | **[0.073, 0.280]** | **[0.052, 0.209]** | **[1.841, 22.991]** | **[0.632, 14.224]** | **[10.677, 13.782]** | **[10.348, 13.617]** |

**Qualitative Study.** To obtain further insights into the quantitative improvements brought about by continuous-time particle filtering, we conduct two qualitative studies on CLPF models using data simulated from the continuous auto-regressive process (CAR), where we observe the most significant improvements.

In Figure 3, we visualize the weights of latent trajectories (i.e., samples/particles) over time without (Figure 3a; equivalent to IWAE) and with (Figure 3b; ours) the advantages of particle-based resampling. The effective sample sizes in both cases are compared in Figure 3c. Both experiments use the exact same sequence of observations as inputs. More transparent and grey segments indicate smaller weights and solid yellow segments indicate larger weights, demonstrating that particle weights based on particle filtering are less skewed and do not decay as much over time. We can also observe particles with small weights getting dropped at the resampling steps (e.g., at time $t = 2.0$), resulting in discontinued trajectories in Figure 3b. Our comparisons underpin the better sampling efficiency of particle filtering compared to an IWAE approximation of the likelihood.

In Figure 4, we compare continuous trajectories extrapolated into the future (blue), conditioned on discrete past observations (red dots). We sample the latent process using two different approaches: (1) by fully relying on the learned variational posterior process, as is usual in latent variable model inference (Figure 4a); and (2) by leveraging our particle filtering approach with the resampling step (Figure 4b). We also include data directly simulated from the ground truth process (red triangles) in both sub-figures and compare them with extrapolated trajectory samples (blues lines) and the average trajectory over these samples (orange lines). The comparison confirms that the average trajectory generated using particle filtering is closer to the simulated samples from the ground truth process and makes better long-term future predictions.

## 4.3 Real-world Data Results

We also evaluate our particle filtering inference approach using models pre-trained on two real-world datasets: Beijing Air Quality Dataset (BAQD) (Zhang et al., 2017) and PTB Diagnostic Database (Bousseljot et al., 1995). Latent SDE and two different variants of CLPF models are considered, CLPF-ANODE and CLPF-iRes, which use a generative variant of augmented neural ODEs (Deng et al., 2020; Dupont et al., 2019) and indexed residual flows (Cornish et al., 2020) to implement their indexed normalizing flows, respectively. The data

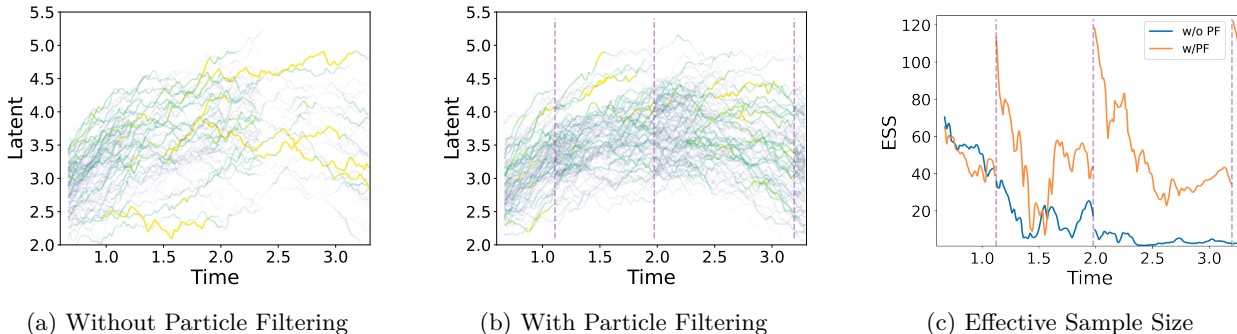

(a) Without Particle Filtering  (b) With Particle Filtering  (c) Effective Sample Size

Figure 3: **Qualitative Evaluation (Particle Weights and Effective Sampling Size).** We show a comparison between the weights of latent trajectories in a CAR process without (left) and with (center) particle filtering. Transparency and color are used to encode the weights. More transparent segments indicate smaller weights than less transparent segments. Yellow indicates larger weights than green than grey. Dashed purple vertical lines indicate particle resampling. The resampling step of our continuous-time particle filtering approach prevents the decay of particle weights. The right figure shows the change of effective sample size of the set of latent samples with and without particle filtering. The discontinuities of the effective sample size trajectories are caused by weights updates with new observations and particle resampling (with particle filtering only).

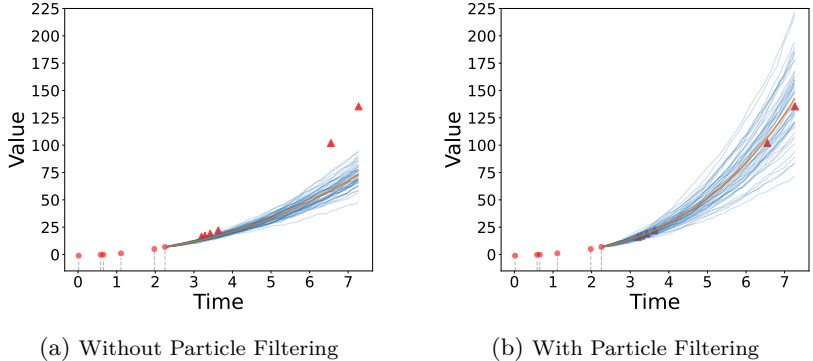

(a) Without Particle Filtering  (b) With Particle Filtering

Figure 4: **Qualitative Evaluation (Approximation Accuracy).** We show extrapolated trajectories (blue) conditioned on a CAR observation sequence (red dots) using a variational posterior (left) and a particle filtering approach (right). The orange trajectory shows the average of all blue trajectories. Simulated realizations (red triangles) from the ground truth process demonstrate the superior accuracy of particle filtering.

sequences are on regular time grids without interpolation of the original data to reflect more realistic settings. In the likelihood estimation task, we compare a particle filtering estimate of the negative log-likelihood against a standard IWAE approximation. A burn-in period similar to standard practice in Markov chain Monte-Carlo methods (Geyer, 2011) is applied to the initial steps of particle filtering, with particle resampling disabled. We hypothesize that this helps the particle filter to begin resampling with a particle set of better quality as the initial few observations might not be informative about the true posterior. Our results are presented in Table 3 and confirm the competitive performance of particle filering on real-world data, which outperforms an IWAE approximation in 4 out of 6 settings.

In the sequential prediction task, we apply particle filtering, sequential importance sampling, and a variational approximation to the pre-trained models and compare their inference performance on a challenging ten-step-ahead prediction. The mean L2 distance between the predicted results and ground truth values across all 10 steps, including the 25th and 75th percentiles, are shown in Table 4. Our results demonstrate that the predictions obtained using particle filtering outperform those based on a variational approximation and sequential importance sampling across all settings and metrics (mean, 25th percentile, and 75th percentile). In Figure 5, we also plot the mean L2 distance of each step for the CLPF-iRes model. We observe that the gap between the particle filtering results and the results obtained with other inference methods is larger for predictions further into the future. Similar trends can be observed for CLPF-ANODE and latent SDE models (see supplementary material).

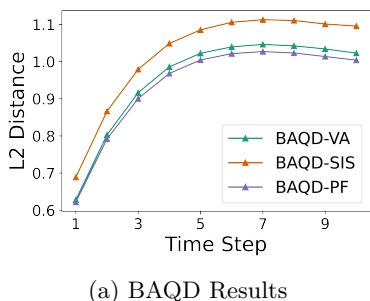 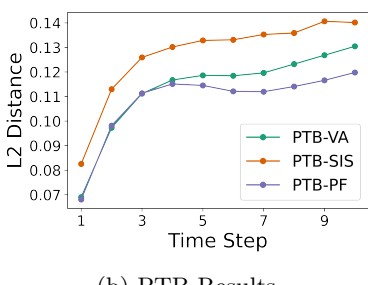

(a) BAQD Results  (b) PTB Results

Figure 5: **Multi-step Sequential Prediction Results of CLPF-iRes on Real-world Datasets.** We show the average L2-distance between predictions and ground truth values over 10 future steps for the CLPF-iRes model on BAQD (left) and PTB (right).

Table 3: **Likelihood Estimation on Real-World Datasets**. We compare the estimated negative log-likelihoods (NLLs) of the proposed particle filtering approach and a standard IWAE approximation on BAQD and PTB, with resampling enabled after step 40. We report the mean and standard deviation across five runs.

| Model (Inference) | BAQD | PTB |
|---|---|---|
| CLPF-ANODE (IWAE) | -0.292±0.001 | **-1.088±0.001** |
| CLPF-ANODE (PF) | **-0.293±0.001** | -1.080±0.005 |
| CLPF-iRes (IWAE) | -0.155±0.001 | **-1.069±0.003** |
| CLPF-iRes (PF) | **-0.166±0.001** | -1.054±0.008 |
| Latent SDE (IWAE) | 1.491±0.004 | -1.280±0.003 |
| Latent SDE (PF) | **1.464±0.003** | **-1.286±0.003** |

Table 4: **Sequential Prediction on Real-World Datasets**. We report the average L2-distance between predictions and ground truth observations over 10 future steps on BAQD and PTB. Results are reported in the format *mean [25th percentile, 75th percentile]*.

| Model (Inference) | BAQD | PTB |
|---|---|---|
| CLPF-ANODE (VA) | 0.904 [0.456, 1.080] | 0.122 [0.034, 0.150] |
| CLPF-ANODE (SIS) | 0.943 [0.470, 1.143] | 0.122 [0.029, 0.142] |
| CLPF-ANODE (PF) | **0.896 [0.443, 1.072]** | **0.107 [0.025, 0.127]** |
| CLPF-iRes (VA) | 0.950 [0.549, 1.155] | 0.116 [0.031, 0.135] |
| CLPF-iRes (SIS) | 1.015 [0.552, 1.258] | 0.126 [0.031, 0.146] |
| CLPF-iRes (PF) | **0.934 [0.519, 1.134]** | **0.107 [0.026, 0.124]** |
| Latent SDE (VA) | 0.968 [0.538, 1.204] | 0.117 [0.043, 0.150] |
| Latent SDE (SIS) | 1.119 [0.612, 1.414] | 0.131 [0.039, 0.168] |
| Latent SDE (PF) | **0.920 [0.515, 1.148]** | **0.102 [0.030, 0.128]** |

## 5 Related Work

As a particle-based inference technique for latent SDEs, our approach is most closely related to previous works in the area of neural differential equations and particle filtering.

**Neural Differential Equations.** The introduction of neural ordinary differential equations (neural ODEs; Chen et al. (2018)) has ignited a new area of research and created interesting questions related to the training and inference of neural differential equations. Chronologically, neural architectures leveraging *ordinary* differential equations predate their stochastic siblings: ODE-RNNs (Rubanova et al., 2019) model the hidden dynamics between consecutive RNN steps with ODEs to reflect non-uniform intervals between observations. A drawback of ODEs is that their solutions depend solely on an initial value, a limitation that has been addressed with neural controlled differential equations (neural CDEs; Kidger et al. (2020)) and neural rough differential equations (neural RDEs; Morrill et al. (2021)), enabling a flexible adaptation to new data. In addition to data-dependent responses, neural ODE processes (NDPs; Norcliffe et al. (2021)) define a distribution over neural ODEs and allow reasoning about the uncertainty associated with latent dynamics. In the context of continuous normalizing flows, Deng et al. (2020) propose a differential deformation of a Wiener base process driven by neural ODEs. The case of sporadic observations in multivariate time-series has been addressed with a continuous-time variant of the gated recurrent unit (GRU; Cho et al. (2014)) and associated Bayesian updates (GRU-ODE-Bayes; De Brouwer et al. (2019)).

The continuous-time particle filtering approach proposed in this paper is most directly applicable to neural *stochastic* differential equations: Tzen & Raginsky (2019) view neural SDEs as the diffusion limit of deep latent Gaussian models (DLGMs; Rezende et al. (2014)) and leverage Girsanov reparameterization to derive a mean-field approximation for neural SDEs. Gradient-based optimization in this framework depends on computationally expensive forward simulations of SDEs, a drawback that has been addressed with the scalable gradients of the stochastic adjoint sensitivity method (Li et al., 2020). Continuous Latent Process Flows (CLPFs; Deng et al. (2021)) leverage latent SDEs for continuous-time indexing of a time-dependent flow

decoder and introduce a piecewise construction of the variational posterior process. Kidger et al. (2021) formalize the insight that neural SDEs and Wasserstein GANs both transform noise into data and express traditional SDE training as a special case of a learnt discriminator statistic.

**Particle Filtering.** Introduced in a seminal work by Gordon et al. (1993) and a quasi-successor to sequential importance sampling (SIS) and sampling and importance resampling (SIR; Rubin (1987)), particle filtering has found applications in numerous inference tasks with intractable state space. Initially viewed as an alternative to extended/unscented Kalman filtering (EKF/UKF; Jazwinski (1970); Julier & Uhlmann (1997)) of non-linear/non-Gaussian dynamical systems, it has since been generalized to particle representations of variables (Koller et al., 1999) or messages (Sudderth et al., 2003) in more general probabilistic graphical models (Koller & Friedman, 2009), and utilized in training neural networks (de Freitas et al., 2000). A finite-sample analysis using concentration bounds and a derivation of convergence rates for such systems can be found in Ihler & McAllester (2009). Using EKF/UKF approximations as the proposal distributions for particle filters has been explored in van der Merwe et al. (2000), as well as extensions for marginalizing out variables in high-dimensional spaces (Doucet et al., 2000). Applications of particle filters to continuous-time models exist in different forms with similar intuition, but they are usually limited to traditional methods without deep architecture: Ng et al. (2005) describe particle filtering in a hybrid-state process in which a discrete state variable evolves according to a continuous-time Markov Jump process and show applications to state estimation tasks of a Mars rover; Murray & Storkey (2007) construct a continuous-time filtering framework for hemodynamic interactions in the brain; Fearnhead et al. (2010) introduce a particle filtering algorithm with random importance weights for continuous-time stochastic processes when the weights of particles are not explicitly available. Among the existing works, (Bain & Crisan, 2009, Chapter 9) and Sottinen & Särkkä (2008) approach importance weighting in continuous-time particle filtering for SDE-based systems using Girsanov's Theorem, which is similar to our proposed framework. However, both works only assume fixed SDEs and importance weighting processes, while our framework deals with variational posteriors and importance weighting processes that are dynamically parametrized by the observations. Additionally, (Bain & Crisan, 2009, Chapter 9) introduces a particle resampling algorithm to reduce the variance of the number of particle offsprings and Sottinen & Särkkä (2008) mainly focus on applications of continuous-time particle filtering in control and navigation scenarios. In the context of variational inference, particle filtering has been used to leverage sequential structure and obtain better estimates of the marginal likelihood (Maddison et al., 2017). Several approaches (Jonschkowski et al., 2018; Zhu et al., 2020; Corenflos et al., 2021) combining machine learning with particle filtering focus on differentiating through the particle filter.

## 6 Discussion and Conclusion

While continuous-time particle filtering can result in significant performance improvements, there are also limitations that should be addressed in future work. One such limitation is the inference speed of particle filtering, which is slower than techniques without particle filtering. For example, using particle filtering for likelihood estimation on synthetic data is about three to five times slower than an efficient implementation of the IWAE estimator. More detailed inference time comparisons can be found in the supplementary material. Every time the weights of particles are updated with a new observation, the particle filtering algorithm will also check whether a resampling should be triggered, and resample the particles if necessary. Therefore, continuous-time particle filters lose opportunities for parallelization during certain inference steps, such as decoding, and incur larger computational overhead when solving the SDE. Furthermore, applying particle filtering to the *training* of deep sequential models has always been a challenging task as gradients cannot be directly backpropagated through the resampling step. Training latent SDE models using continuous-time particle filtering (as opposed to the inference phase considered in this work) also faces this challenge and would require specialized techniques, such as Gumbel-softmax or policy gradients.

In this work we proposed the adoption of continuous-time particle filtering as a generic drop-in replacement of variational inference methods for latent SDE models. We defined a mathematically rigorous framework for continuous-time particles, including importance weighting and update schemes, and demonstrated how to leverage the proposed framework in two common inference tasks, likelihood evaluation and sequential prediction. The effectiveness and generality of continuous-time particle filtering has been shown on two models in the latent SDE family, CLPF and latent SDE, four continuous-time stochastic processes, and two real-world datasets.

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

## A    Piece-Wise Construction of Posterior Process

This section provides a complete and detailed description of the piece-wise approach to the construction of the posterior process. Our piece-wise construction of the posterior process is based on the following fact of Wiener process: Given a time grid $0 = t_0 < t_1 < \cdots < t_n = T$, we can sample a Wiener process trajectory of length $T$ via sampling from $n$ independent Wiener processes $\{\boldsymbol{W}_t^{(i)}\}_{i=1}^n$, each of length $t_i - t_{i-1}$ defined on the filtered probability space $(\Omega^{(i)}, \mathcal{F}_t^{(i)}, P^{(i)})$ with distribution $P^{(i)}$, in the following way:

$$
\begin{aligned}
&\boldsymbol{W}_t(\omega^{(1)}, \omega^{(2)}, \ldots, \omega^{(n)}) \\
&= \sum_{\{i : t_i < t\}} \boldsymbol{W}_{t_i - t_{i-1}}^{(i)}(\omega^{(i)}) + \boldsymbol{W}_{t - t_{i*}}^{(i^*)}(\omega^{(i^*)})
\end{aligned}
\tag{17}
$$

where $i^* = \max\{i : t_i < t\} + 1$ and $\omega^{(i)} \in \Omega^{(i)}$. This construction of Wiener process allows us to solve the stochastic differential equation

$$
\mathrm{d}\boldsymbol{Z}_t = \mu_\theta(\boldsymbol{Z}_t, t)\,\mathrm{d}t + \sigma_\theta(\boldsymbol{Z}_t, t)\,\mathrm{d}\boldsymbol{W}_t
\tag{18}
$$

and sample $\{\boldsymbol{Z}_{t_i}\}_{i=1}^n$ in a piece-wise manner given $\boldsymbol{Z}_{t_0}$. Taking $\boldsymbol{Z}_{t_{i-1}}$ as the initial condition, we can sample $\boldsymbol{Z}_{t_i}$ by solving the SDE

$$
\mathrm{d}\boldsymbol{Z}_t = \mu_\theta(\boldsymbol{Z}_t, t)\,\mathrm{d}t + \sigma_\theta(\boldsymbol{Z}_t, t)\,\mathrm{d}\boldsymbol{W}_t^{(i)}
\tag{19}
$$

in the interval between $t_{i-1}$ and $t_i$, i.e.

$$
\begin{aligned}
\boldsymbol{Z}_{t_i} = \boldsymbol{Z}_{t_{i-1}} &+ \int_{t_{i-1}}^{t_i} \mu_\theta(\boldsymbol{Z}_s, s)\,\mathrm{d}s \\
&+ \int_{t_{i-1}}^{t_i} \sigma_\theta(\boldsymbol{Z}_s, s)\,\mathrm{d}\boldsymbol{W}_{s - t_{i-1}}^{(i)}.
\end{aligned}
\tag{20}
$$

Therefore we rewrite the expectation on the left-hand side of Eq.(7) in Section 2.1 as

$$
\mathbb{E}_{P^{(1)}}\left[\ldots \mathbb{E}_{P^{(i)}}\left[\ldots \mathbb{E}_{P^{(n)}}\left[f\left(\{\boldsymbol{Z}_{t_i}\}_{i=1}^n\right)\right]\ldots\right]\ldots\right].
\tag{21}
$$

For the simplicity of presentation, we rewrite the term in the expectation $\mathbb{E}_{P^{(i)}}[\cdot]$ as a function of $\boldsymbol{Z}_{t_i}$ conditioned on $\{\boldsymbol{Z}_{t_k}\}_{k=1}^{i-1}$, i.e.,

$$
\begin{aligned}
&\mathbb{E}_{P^{(i)}}\left[\ldots \mathbb{E}_{P^{(n)}}\left[f\left(\{\boldsymbol{Z}_{t_i}\}_{i=1}^n\right)\right]\ldots\right] \\
=&\mathbb{E}_{P^{(i)}}\left[f^{(i)}(\boldsymbol{Z}_{t_i} | \{\boldsymbol{Z}_{t_j}\}_{j=1}^{i-1})\right]
\end{aligned}
\tag{22}
$$

for some function $f^{(i)}(\cdot \,|\, \{\boldsymbol{Z}_{t_j}\}_{j=1}^{i-1})$ thanks to the Markov property of SDE solutions. For each interval $[t_{i-1}, t_i]$, we can define a posterior process SDE

$$
\mathrm{d}\tilde{\boldsymbol{Z}}_t = \mu_{\phi^{(i)}}(\tilde{\boldsymbol{Z}}_t, t)\,\mathrm{d}t + \sigma_\theta(\tilde{\boldsymbol{Z}}_t, t)\,\mathrm{d}\boldsymbol{W}_t^{(i)}
\tag{23}
$$

with $\phi^{(i)}$ potentially being (partially) parameterized by $\{\boldsymbol{Z}_{t_k}\}_{k=1}^{i-1}$ and define distribution $Q^{(i)}$ for $(\Omega^{(i)}, \mathcal{F}_t^{(i)})$ according to Girsanov Theorem (Oksendal, 2013) such that

$$
\begin{aligned}
&\mathbb{E}_{P^{(i)}}\left[f^{(i)}(\boldsymbol{Z}_{t_i} | \{\boldsymbol{Z}_{t_k}\}_{k=1}^{i-1})\right] \\
=&\mathbb{E}_{Q^{(i)} | \{\boldsymbol{z}_{t_k}\}_{k=1}^{i-1}}\left[f^{(i)}(\tilde{\boldsymbol{Z}}_{t_i} | \{\boldsymbol{Z}_{t_k}\}_{k=1}^{i-1})\right] \\
=&\mathbb{E}_{P^{(i)}}\left[f^{(i)}(\tilde{\boldsymbol{Z}}_{t_i} | \{\boldsymbol{Z}_{t_k}\}_{k=1}^{i-1})\boldsymbol{M}^{(i)}\right]
\end{aligned}
\tag{24}
$$

Please refer to Section 3.1 for the details of defining $Q^{(i)}$ and $\boldsymbol{M}^{(i)}$. As a result, Eq. 21 can be rewritten in the following way

$$
\begin{aligned}
&\mathbb{E}_{P^{(1)}}\left[\ldots \mathbb{E}_{P^{(i)}}\left[\ldots \mathbb{E}_{P^{(n)}}\left[f\left(\{\boldsymbol{Z}_{t_i}\}_{i=1}^n\right)\right]\ldots\right]\ldots\right] \\
&= \mathbb{E}_{Q^{(1)}|\{\tilde{\boldsymbol{Z}}_{t_1}\}}\Big[\ldots \mathbb{E}_{Q^{(i)}|\{\tilde{\boldsymbol{Z}}_{t_k}\}_{k=1}^{i-1}}\Big[ \\
&\qquad\ldots \mathbb{E}_{Q^{(n)}|\{\tilde{\boldsymbol{Z}}_{t_k}\}_{k=1}^{n-1}}\left[f(\{\tilde{\boldsymbol{Z}}_{t_i}\}_{i=1}^n)\right]\ldots\Big]\ldots\Big] \\
&= \mathbb{E}_{P^{(1)}}\Big[\ldots \mathbb{E}_{P^{(i)}}\Big[\ldots \mathbb{E}_{P^{(n)}}\big[ \\
&\qquad f(\{\tilde{\boldsymbol{Z}}_{t_i}\}_{i=1}^n)\boldsymbol{M}^{(n)}\big]\ldots\boldsymbol{M}^{(i)}\Big]\ldots\boldsymbol{M}^{(1)}\Big].
\end{aligned}
\tag{25}
$$

For latent SDE model Li et al. (2020), $\phi^{(i)}$ is completely determined by observation sequence $\{\boldsymbol{x}_{t_i}\}_{i=1}^n$ and remains the same for each time interval $[t_{i-1}, t_i]$. In CLPF models Deng et al. (2021), $\phi^{(i)}$ is parameterized by $\{\boldsymbol{x}_{t_k}\}_{k=1}^{i}$ and $\{\hat{\boldsymbol{Z}}_{t_k}\}_{k=1}^{i-1}$ in the interval between $t_{i-1}$ and $t_i$

## B  Synthetic Data Simulation and Training Settings

In our experiments, data sequences are sampled from four common continuous stochastic processes: geometric Brownian motion (GBM), linear SDE (LSDE), continuous auto-regressive process (CAR), and stochastic Lorenz curve (SLC). The sequences are simulated using the Euler-Maruyama method Bayram et al. (2018) with a fixed step size of $1e-5$ in the time interval $[0, 30]$ for GBM, LSDE, and CAR and in the time interval $[0, 2]$ for SLC. Below are the parameters we use for simulation in each process:

**Geometric Brownian Motion.** Observations are sampled from the SDE $\mathrm{d}\boldsymbol{X}_t = 0.2\boldsymbol{X}_t\,\mathrm{d}t + 0.1\boldsymbol{X}_t\,\mathrm{d}\boldsymbol{W}_t$, with an initial value $\boldsymbol{X}_0 = 1$.

**Linear SDE.** We simulate sequences from the the SDE $\mathrm{d}\boldsymbol{X}_t = (0.5\sin(t)\boldsymbol{X}_t + 0.5\cos(t))\,\mathrm{d}t + \frac{0.2}{1+\exp(-t)}\,\mathrm{d}\boldsymbol{W}_t$ with initial value 0.

**Continuous AR(4) Process.** A CAR process $\boldsymbol{X}_t$ can be viewed as the linear projection of a process defined by a high-dimensional SDE to a low-dimensional one. The high-dimensional SDE and linear projection in our fourth-order CAR process are

$$
\begin{aligned}
\mathrm{d}\boldsymbol{Y}_t &= A\boldsymbol{Y}_t\,\mathrm{d}t + e\,\mathrm{d}\boldsymbol{W}_t, \\
\boldsymbol{X}_t &= [1, 0, 0, 0]\boldsymbol{Y}_t,
\end{aligned}
\quad \text{where}
$$

$$
A = \begin{bmatrix} 0 & 1 & 0 & 0 \\ 0 & 0 & 1 & 0 \\ 0 & 0 & 0 & 1 \\ 0.002 & 0.005 & -0.003 & -0.002 \end{bmatrix} \text{ and } e = [0, 0, 0, 1].
\tag{26}
$$

**Stochastic Lorenz Curve.** The stochastic Lorenz curve is defined by the following three-dimensional SDE

$$
\begin{aligned}
\mathrm{d}\boldsymbol{X}_t &= 10(\boldsymbol{Y}_t - \boldsymbol{X}_t)\,\mathrm{d}t + 0.1\,\mathrm{d}\boldsymbol{W}_t, \\
\mathrm{d}\boldsymbol{Y}_t &= (\boldsymbol{X}_t(28 - \boldsymbol{Z}_t) - \boldsymbol{Y}_t)\,\mathrm{d}t + 0.28\,\mathrm{d}\boldsymbol{W}_t, \\
\mathrm{d}\boldsymbol{Z}_t &= (\boldsymbol{X}_t\boldsymbol{Y}_t - \frac{8}{3}\boldsymbol{Z}_t)\,\mathrm{d}t + 0.3\,\mathrm{d}\boldsymbol{W}_t.
\end{aligned}
\tag{27}
$$

7000 sequences are simulated for training and 1000 sequences for validation. The observation time stamps of training data are sampled from homogeneous Poisson processes with intensity values of $\lambda = 2$ for geometric Brownian motion, linear SDE, and continuous auto-regressive process and $\lambda = 20$ for stochastic Lorenz curve.

The experiment settings and model architectures are aligned with the synthetic data experiment settings of CLPF Deng et al. (2021). For both CLPF and latent SDE models, we optimize the IWAE bound of negative log likelihood per step estimated by 3 latent samples with a batch size of 128 and learning rate of 0.001.

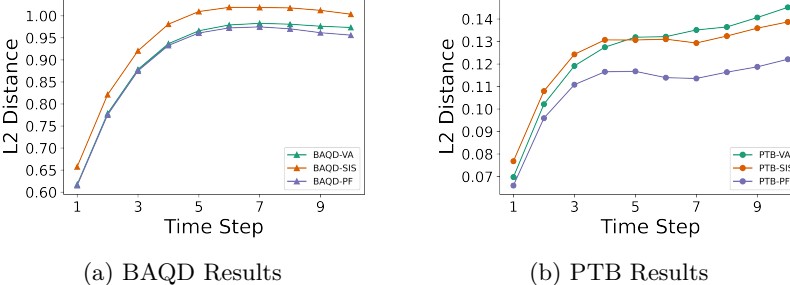

(a) BAQD Results         (b) PTB Results

Figure 6: **Multi-step Sequential Prediction Results of CLPF-ANODE on Real-world Datasets.** We show the average L2-distance between predictions and ground truth values over 10 future steps for the CLPF-ANODE model on BAQD (left) and PTB (right).

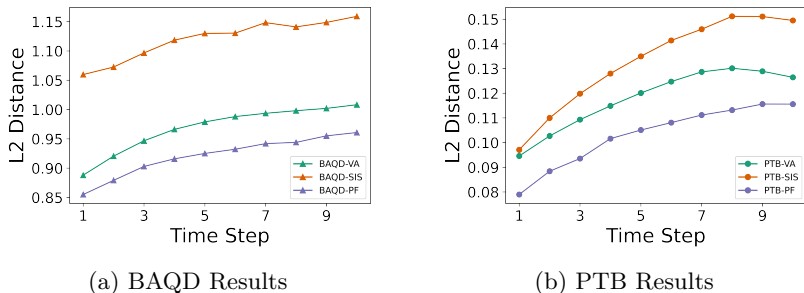

(a) BAQD Results         (b) PTB Results

Figure 7: **Multi-step Sequential Prediction Results of Latent SDE on Real-world Datasets.** We show the average L2-distance between predictions and ground truth values over 10 future steps for the latent SDE model on BAQD (left) and PTB (right).

## C  Real-world Data Experiment Settings

We use two real-world datasets for evaluating our particle filtering inference method: 1) Beijing Air Quality Dataset (BAQD; Zhang et al. (2017)) and 2) PTB Diagnostic Database (PTB; Bousseljot et al. (1995)). We use the temperature, pressure and wind speed dimension of the BAQD dataset. The data are recorded at the rate of once per hour and we excerpt the original data into sequences of length of 168 covering the records of a whole week. The BAQD data are normalized dimension-wise. For the PTB dataset, we use the one-dimensional ambulatory electrocardiogram recordings recorded at the frequency of 125 Hz with a maximum sequence length of 650. The indices of the sequences are treated as real numbers and rescaled with shift into the time interval of [0, 30] for BAQD dataset and the interval of [0, 120] for PTB dataset with s. Following the practice of CLPF Deng et al. (2021), we also use asynchrounous sequences for training on real-world datasets: The observation time stamps are also sampled from homogeneous Poisson point process with $\lambda = 2$ and the value of the closest observation available after rescaling the indices are taken as the observation of the sampled observation time stamp. We also optimize the model using IWAE bound with 5 latent samples and a training batch size of 25. On likelihood evaluation and sequential prediction tasks, we use synchronous data obtained by subsampling the sequence with real-valued time stamps $\{(t_i, \boldsymbol{x}_{t_i})\}_{i=1}^n$ with a rate of 2 for BAQD dataset and a rate of 3 for PTB dataset.

## D  Additional Real-World Experiment Results

We present additional sequential prediction results for CLPF-ANODE and latent SDE models in Figure 6 and Figure 7, respectively. Both confirm previously observed trends: particle filtering performs better than a variational approximation and sequential importance sampling, and in most cases the gap between particle filtering and the other methods is larger at the later steps than at the initial steps.

# E   Comparison of Inference Time

Table 5 presents the comparison of the wall-clock inference time reported in seconds between the particle-filtering-based inference method and IWAE Burda et al. (2015) methods with efficient implementation for CLPF Deng et al. (2021) and latent SDE Li et al. (2020) models on likelihood esimtation tasks based on 10 batches of data for each simulated process. The batch size is 10 and the number of latent samples is 125.

Table 5: **Comparison of Computational Time between Particle Filtering and IWAE on Likelihood Estimation**. The value of $\lambda$ in the parenthesis of each processes indicates the intensity of the poisson process from which observation time stamps are sampled from.

| Model (Inference) | GBM ($\lambda = 2$) | LSDE ($\lambda = 2$) | CAR ($\lambda = 2$) | SLC ($\lambda = 20$) |
|---|---|---|---|---|
| Latent SDE (IWAE) | 25.615 | 35.310 | 12.861 | 8.013 |
| Latent SDE (PF) | 71.008 | 72.336 | 35.219 | 25.376 |
| CLPF (IWAE) | 89.076 | 89.083 | 86.838 | 98.457 |
| CLPF (PF) | 429.740 | 471.378 | 349.328 | 614.471 |

## F  Pseudo Code for Continuous-Time Particle Filtering

Algorithm 1 presents a summarized version of the continuous-time particle filtering algorithm in pseudocode.

---

**Algorithm 1:** Continuous-Time Particle Filter

**input:**   Observation sequence with time points $\{(t_i, \boldsymbol{x}_{t_i})\}_{i=1}^{n}$;
Latent process drift function with adaptable parameters
$\mu(z, t; \text{parameters}) : \mathbb{R}^M \times \mathbb{R} \to \mathbb{R}^M$;
Latent process variance function $\sigma(z, t) : \mathbb{R}^M \times \mathbb{R} \to \mathbb{R}^M \times \mathbb{R}^M$;
Parameters for the drift function $\mu$ in the original prior process of latent SDE $\theta$;
A function generating parameters for drift function from sequence parameters $PARAM\_GEN(\{\boldsymbol{z}_{t_k}\}_{k=1}^{i})$;
A sampler of an $M$-dimensional Wiener process trajectory, with time length $t$ as input $SAMPLER(t)$;
Conditional likelihood evaluation function of observations $p(\boldsymbol{x}_{t_i} | \{\boldsymbol{z}_{t_k}\}_{k=1}^{i}, \{\boldsymbol{x}_{t_k}\}_{k=1}^{i-1})$;
Initial state of the latent process $\boldsymbol{z}_{t_0}$;
Number of particles $N$ ;
A boolean resampling condition function
$RESAMPLE\_CON(\{\boldsymbol{w}_j\}_{j=1}^{N})$;

**output:** A set of particles with weights $\{(\{\omega_j^{(k)}\}_{k=1}^{n}, \boldsymbol{w}_j^{(n)})\}_{j=1}^{N}$;

**Initialization** : $\boldsymbol{w}_j^{(0)} \leftarrow \frac{1}{N}, \boldsymbol{z}_{t_0,j} = \boldsymbol{z}_{t_0}$ **for** $j \leftarrow 1$ **to** $N$; particle_set $\leftarrow \{(\{\}, \boldsymbol{w}_j^{(0)})\}_{j=1}^{N}$ ;

**1** **for** $i \leftarrow 1$ **to** $n$ **do**

**2**    **for** $j \leftarrow 1$ **to** $N$ **do**

**3**       $\omega_j^{(i)} \leftarrow SAMPLER(t_i - t_{i-1})$;

**4**       $\phi_j^{(i)} \leftarrow PARAM\_GEN(\{\boldsymbol{z}_{t_i,j}\}_{k=0}^{i-1})$;

      /* Solve the stochastic differential equation in the interval $[t_{i-1}, t_i]$ given the Wiener process path sample $\omega_j^{(i)}$.                                                                                    */

**5**       $\boldsymbol{M}_j^{(i)}, \boldsymbol{z}_{t_i,j} \leftarrow AUG\_SDESOLVE(t_{i-1}, t_i, \boldsymbol{z}_{t_{i-1}}, \omega_j^{(i)}, \sigma(\cdot, \cdot), \mu(\cdot, \cdot, \theta), \mu(\cdot, \cdot, \phi_j^{(i)}))$;

      /* $AUG\_SDESOLVE$ not only solves the SDE but also computes the importance weights of sample between the prior and proposal distributions.                                             */

**6**       $\{\omega_j^{(k)}\}_{k=1}^{i} \leftarrow CONCAT(\{\omega_j^{(k)}\}_{k=1}^{i-1}, \{\omega_j^{(i)}\})$;

**7**       $\tilde{\boldsymbol{w}}_j^{(i)} \leftarrow \boldsymbol{w}_j^{(i)} p(\boldsymbol{x}_{t_i} | \{\boldsymbol{z}_{t_k,j}\}_{k=1}^{i}, \{\boldsymbol{x}_{t_k}\}_{k=1}^{i-1}) \boldsymbol{M}_j^{(i)}$;

**8**    **end**

   /* Normalize the weights.                                                                                                       */

**9**    $\boldsymbol{w}_j^{(i)} \leftarrow \frac{\tilde{\boldsymbol{w}}_j^{(i)}}{\sum_{j=1}^{N} \tilde{\boldsymbol{w}}_j^{(i)}}$ **for** $j \leftarrow 1$ **to** $N$;

**10**   **if** $RESAMPLE\_CON(\{\omega_j^{(i)}\}_{j=1}^{N})$ **then**

      /* Resample the particles from categorical distributions defined by the importance weights and reset the weights.                                                                     */

      $\{\{\omega_j^{(k)}\}_{k=1}^{i}\}_{j=1}^{N} \leftarrow CAT\_SAMPLE(\{\{\omega_j^{(k)}\}_{k=1}^{i}\}_{j=1}^{N}, \{\boldsymbol{w}_j^{(i)}\}_{j=1}^{N})$;

**11**      $\boldsymbol{w}_j^{(i)} \leftarrow \frac{1}{N}$ **for** $j \leftarrow 1$ **to** $N$;

**12**   **end**

**13**   particle_set $\leftarrow \{(\{\omega_j^{(k)}\}_{k=1}^{i}, \boldsymbol{w}_j^{(i)})\}_{j=1}^{N}$;

**14** **end**

**Return:** particle_set

---

