# OpenReview forum: "Continuous-time Particle Filtering for Latent Stochastic Differential Equations"
_TMLR — Rejected by TMLR_

### Review · Reviewer_nVVR · 2024-05-13

**Summary Of Contributions:**

The authors propose to use continuous time particle filters during inference in neural architectures with latent SDEs. In the experiments, the authors use the proposed method for likelihood estimation and sequential prediction tasks, and show its effectiveness on multiple datasets.

**Audience:**

Yes

**Claims And Evidence:**

Yes

**Requested Changes:**

The paper seems to contain good technical points, but the presentation of the paper is not very good.
The authors would be better to add the following points to your manuscript in a clear and concise manner.
- What is the problem setting to be addressed?
- Why is it important? What can it be used for?
- Why couldn't it be solved with conventional methods?
- What is the idea behind the proposed method?

Before going into technical details, it would be good to add an intuitive diagram and explanation that illustrates the idea of the proposed method.

Does the proposed method require solving SDEs? Is the Ito integral assumed in this study, or does the Stratonovich integral give similar results?

**Strengths And Weaknesses:**

Strengths
- This paper is technically solid.
- The evaluation is conducted using synthetic and real-world datasets.

Weaknesses
- This paper is not readable for non-experts.
- The problem setting, the novelty, the technical ideas, the significance, etc. are not well-written in the manuscript, easy-to-understand manner.

---

> ### Author Response · Authors · 2024-06-11
> **Response to Reviewer nVVR**
>
> We appreciate the reviewer’s valuable suggestions and have addressed them in our revised manuscript; see our reply to all reviewers for details. Below we answer the remaining questions.
>
> **[Question 1 — Problem Setting]**
>
> The main focus of our work is inference in pre-trained neural latent stochastic differential equation models. Specifically, we consider inference tasks that require estimating the posterior distribution conditioned on a sequence of irregularly-spaced observations at real-valued points in time — such as likelihood estimation or sequential prediction in a filtering setting as described in Section 3.2. A neural latent stochastic differential equation model in our problem setting is a generative model consisting of a prior latent process, a neural decoder that decodes the latent process into observations, and a variational posterior process that approximates the posterior process conditioned on the observations; see Section 2 for details on this setup.
>
> In our revised manuscript we have added an overview figure (Figure 1) that provides a high-level description of our framework’s components and their interactions.
>
>
> **[Questions 2/3 — Drawbacks of Existing Approaches and Benefits of the Proposed Approach]**
>
> Existing inference approaches for neural latent SDEs rely on variational approximations or sequential importance weighting. In case of variational approximations, there is no guarantee how well the variational posterior process can approximate the true posterior process. As the example in Figure 3(a) shows, the trajectories sampled from the variational posterior process are drifting away from the realizations simulated from the ground truth process, because the variational posterior process cannot capture the real posterior distribution of the latent process conditioned on past observations. Sequential importance weighting, on the other hand, is prone to concentrating the particle weights on a few samples. This scenario is shown in Figure 2(a) and leads to lower effective sample size and sample efficiency.
>
> Our proposed approach builds on the particle filtering framework, which is known as an effective Sequential Monte-Carlo technique. The benefits of our proposed approach include better sample efficiency (by resampling the latent state samples) and asymptotic convergence to the true posterior distribution as the number of latent state samples increases. Both properties lead to improved results in inference tasks, as we show in Figure 3(b) and Figure 2(b).
>
> In our revised manuscript we have reworked our discussion of existing inference techniques and the role of particle filtering in the introduction.
>
> **[Question 4 — Core Idea]**
>
> The core idea of the proposed approach is to represent the posterior distribution of the latent state with a set of weighted particles. It consists of three steps: (1) representing particles by a sequence of Wiener process trajectory segments, such that the increment of each segment is itself a Wiener process trajectory; (2) using Girsanov’s theorem to assign importance weights to the segments in each particle and normalizing the weights; and (3) applying boot-strap resampling to the particles when the effective sample size decays. The technical details for these three steps are discussed in Section 3.1.
>
> **[Question 5 — Solving SDEs / Ito Integral vs. Stratonovich Integral]**
> Our approach requires numeric solutions of SDEs and stochastic integrals. We assume the Ito integral throughout this work and have clarified this fact in Section 2. However, it is worth mentioning that an SDE defined using the Stratonovich integral can be converted to an SDE defined using the Ito integral with the same solution, and vice versa [1, Chapter 3.3]. Such a conversion will only impact the drift term of the stochastic differential equation and not the computation of the importance weights as long as Novikov’s condition (Equation 4) is met. Furthermore, we note that importance weighting in our work is applied to the samples of the Wiener process and is thus independent of how the stochastic integral is defined in terms of Wiener process samples.
>
> **[References]**
>
> [1] Bernt Oksendal: “Stochastic differential equations: an introduction with applications.” Springer Science & Business Media, 2013.

---

### Review · Reviewer_L43e · 2024-05-29

**Summary Of Contributions:**

The authors propose a method for performing Monte-Carlo estimation of functions of discrete observations of continuous time stochastic differential equations.

They apply this method to inference of the latent path of a stochastic differential equation in a learnt latent stochastic differential process model.

**Audience:**

Yes

**Broader Impact Concerns:**

None.

**Claims And Evidence:**

Yes

**Requested Changes:**

- Mainly I am concerned with addressing the weaknesses. I would like to see the following:
  - A better background on particle filtering to make the paper more readable.
  - A better distinction on where the novelty of the paper lies.
  - A better analysis of the proposed particle filter outside the ML context, perhaps with the suggested experiment.
  - A tightening up of the earlier experiments with error bars reported over training seeds of the machine learning models used.

**Strengths And Weaknesses:**

Strengths:
- The idea is clean and neat. The core concepts of the idea are well presented.
- I am confident that the approach proposed is sound.

Weaknesses:
- For a paper about particle filtering, the authors provide very little in the way of a background on what this is. I think readability of the paper would benefit greatly from a brief introduction to particle filtering, the basic algorithm, and when it is appropriate to use.
- The paper is not very clear at delineating where novelty is coming in. Parts of section 3, for example the construction of piecewise importance weights, are drawn directly from [1], and the Girsanov approach to computing the importance weights goes back to [2].
- There is a tension in the paper between developing this new particle filtering approach, and the application to the machine learning task. The novelty is all in the development of the particle filtering method, but it is only applied and discussed in the context of learnt latent SDE models. The paper would benefit from stronger investigation of the particle filter outside the ML setting on more tasks where the performance of the particle filter is the only aspect in question, for example using the synthetic data tasks, but simply using the known latent SDE terms, rather than trying to learn them.
- The experiments are quite weak. Tables 1 and 2 contain no error bars, and the improvements over baseline methods seem to be slight. The experiments relating to tables 3 and 4 are more convincing. Generally in all the experiments multiple runs over training seeds of the machine learning models seem to be missing


[1] https://arxiv.org/pdf/2106.15580
[2] Särkkä, Simo, and Tommi Sottinen. "Application of Girsanov Theorem to Particle Filtering of Discretely Observed Continuous-Time Non-Linear Systems." Bayesian Analysis 3.3 (2008)

---

> ### Author Response · Authors · 2024-06-11
> **Response to Reviewer L43e**
>
> We thank the reviewer for suggesting the inclusion of a gentle introduction to particle filtering, which we have added as Section 2.1 in our revised manuscript. We also agree with the benefits of multiple runs / error bars and an experiment on non-neural SDEs with known dynamics. These experiments are currently running and will be added to the manuscript in a follow-up. Below we comment on the reviewer’s remaining questions.
>
> **[Distinction with Existing Works]**
>
> The novelty of our work lies in the application of continuous-time particle filtering to inference tasks in latent neural SDEs. In our response to reviewer xVMb we further discussed the challenges of bridging the gap between existing continuous-time particle filtering frameworks and their application in the context of latent neural stochastic differential equations. We acknowledge that our contributions are not fully decoupled from some existing works (piece-wise Wiener process, importance weighting using Girsanov’s theorem). However, they are essential parts of our continuous-time particle filtering approach as we take the Wiener segments in each time interval as particles and perform an importance weighting in each update step. Therefore we include them in our approach section but have ensured that the relevant literature is properly cited.
>
> In addition to the particle filtering framework itself, we also show how inference tasks like likelihood evaluation and sequential prediction can be cast as expectations over the true posterior of the latent process and describe how the proposed particle filter can be used in these cases.

---

> ### Author Response · Authors · 2024-06-13
> **Updates on Response to Reviewer L43e**
>
> **[Continuous-time Particle Filtering Outside of Neural SDEs]**
>
> We agree with the reviewer on the potential of continuous-time particle filtering outside of neural SDEs. However, such applications come with their own challenges. For example, in an architecture with fixed (non-learnable) variational posterior process $\tilde{\boldsymbol{Z}}_t$, an appropriate (constant) drift term $\mu$ must be selected. The choice of $\mu$, in turn, has a significant impact on the effectiveness of the particle filter.
>
> In a preliminary investigation, we set $\mu = 0.5$ and obtained the following results in the sequential prediction task for the Stochastic Lorenz Curve ($\lambda=20$):
>
> - Mean error: 25.180
> - [25th, 75th] percentile: [16.133, 30.163]
>
> These results are considerably worse than their counterparts with a learned variational posterior (Table 2) and reinforce our believe in the strength of an observation-dependent drift term parametrized by a deep network, which makes it a better approximation of the true posterior process. Allowing for dynamic parametrizations is an important element of this work and further discussed in our response to Reviewer xVMb.
>
> **[Error Bars for Table 1 and Table 2]**
>
> We have updated Table 1 with mean and standard deviation values across multiple runs and Table 2 with 25th and 75th percentiles to align them with Table 3 and Table 4 in the revised version.

---

### Review · Reviewer_xVMb · 2024-05-30

**Summary Of Contributions:**

This paper provides a methodology for particle filtering for the class of models called latent stochastic differential equations (related to neural stochastic differential equations). As a continuous time approach, the paper uses a proposal and computes weights via a change of measure. Several experiments are presented on various models.

**Audience:**

Yes

**Claims And Evidence:**

Yes

**Requested Changes:**

Detailed comments:

1) In my opinion, the paper's contribution is not novel. The idea is not novel and explored massively in the literature, see some references:

- Random-weight particle filtering of continuous time processes

in fact the idea is so well known, it is in textbooks - see Fundamentals of Stochastic Filtering, Bain and Crisan.

I suggest authors to significantly update their literature review with at least important relevant works from this field.

2) It seems that the idea authors try to develop is to use these techniques (which are not novel) for latent stochastic differential equations. However, this is not very novel - as every other continuous time process, particle filtering of course can be applied here. Can authors clarify what the specific contributions here are?

3) The real challenge in this setting is the existence of a parameter of (several) neural networks $\theta$ - both in the latent dynamics and in the decoder. The estimation of this can be done using many techniques also from parameter estimation (in state-space models) literature. It is not clear in the paper if these parameters were fitted or not, please clarify.

4) Please compute and report effective sample size (ESS). It is cited that resampling is only done when "When certain criteria are met" but it is important to clarify it. Plot ESS over time.

In general, as I made the point a few times, I really do not see what the novelty in this paper is; given the missing large continuous-time filtering literature, lack of rigorous evaluation from Monte Carlo perspective, also lack of parameter estimation techniques.

**Strengths And Weaknesses:**

While the relevance and popularity of these models make the contribution relevant, I think there are two core weaknesses:

1) A big literature on continuous-time filtering (and particle filtering) is missed, using the continuous time representation of SDEs to compute weights is a very old idea. There are so many papers on this topic, it is easy to find many of them - some of them will be cited in my more detailed comments.
2) Given P and Q, the filtering is done - but this is a very traditional topic. The challenge is to fit $\theta$ in this setting, which is skipped.

---

> ### Author Response · Authors · 2024-06-11
> **Response to Reviewer xVMb**
>
> **[Novelty]**
>
> We thank the reviewer for his insightful comments and constructive feedback. We acknowledge the existence of continuous-time particle filters for stochastic differential equations as well as the segment-wise approach toward the Wiener process; both are important prerequisites for our work. We thank the reviewer for pointing us to additional resources and have expanded our previous discussion on these topics [1, 2] with the new references suggested by the reviewer [3, 4] in our revised manuscript. We also make use of some core elements of generic particle filtering, such as weight normalization and particle resampling, and have included an introduction to the field in Section 2.1 to improve accessibility.
>
> Our work demonstrates the effectiveness of adopting continuous-time particle filtering as a drop-in replacement for inference in neural latent stochastic differential equations. To the best of our knowledge, the proposed framework is the first to provide a particle filtering-based alternative to the widespread use of variational approximations or sequential importance weighting for this class of models. We argue that our specific contributions of adapting continuous-time particle filtering for latent neural SDE models are non-trivial due to the following challenges:
>
> 1. The generic formulation of continuous-time particle filtering usually assumes that the pair of processes $\boldsymbol{Z}_t$ and $\tilde{\boldsymbol{Z}_t}$ for inducing importance weights, or the importance weight process $M_t$, is given. However, in the context of latent neural SDEs, the parametrization of $\tilde{\boldsymbol{Z}_t}$ and the corresponding distribution $Q$ are not unique, leading to different importance weighting processes $M_t$ for different $\tilde{\boldsymbol{Z}_t}$. Typically, $\tilde{\boldsymbol{Z}_t}$ is parametrized differently by the observations for each sample. Given the non-uniqueness of $Q$ and $\tilde{\boldsymbol{Z}_t}$, we need to refer to $Q$ as instances from a family of distributions induced by different parametrizations of $\tilde{\boldsymbol{Z}_t}$ with the same marginal distributions. This is shown in Equation (7) and Equation (11).
> 2. The $\tilde{\boldsymbol{Z}_t}$ in the generic formulation of continuous-time particle filtering is usually assumed to be defined by a single SDE. This is also the approach taken by the vanilla latent SDE model [5, 6]. However, in the general latent neural SDE setting, the $\tilde{\boldsymbol{Z}_t}$ can be more flexible. For example, CLPF [2] takes a segment-wise approach towards the construction of $\tilde{\boldsymbol{Z}_t}$, with the stochastic dynamics of $\tilde{\boldsymbol{Z}_t}$ in the interval of $[t\_{i-1}, t_i]$ parameterized by $\tilde{\boldsymbol{z}\_{t_j}}$, for j <= i-1. To adapt continuous-time particle filtering for both vanilla latent SDE [5, 6] and CLPF [2], we factorize both $P$ and $Q$ into a product of probability distributions and apply importance weighting individually for each time interval between observations.
>
> **[Parameter Learning]**
>
> A key challenge in using particle filtering for parameter learning arises from the difficulty of backpropagating through the particle resampling step, which is non-differentiable. Various approaches for backpropagating through the resampling step in particle filters based on a Gumbel softmax or policy gradient have been studied. Parameter learning for latent neural SDE models based on particle filtering would be a direct application of these works to our proposed framework. Our contribution of bridging the gap between generic continuous-time particle filtering and the architectural specifics of neural latent SDE models are crucial for such applications. Considering the nature of particle filtering as a method for estimating the internal states of a dynamic system, we believe showing inference results on pre-trained latent neural SDE models validates our core contributions.
>
> **[Experiments]**
>
> The proposed ESS plot will be added in a follow-up.
>
> **[References]**
>
> [1] T.Sottinen and S.Särkkä: “Application of Girsanov theorem to particle filtering of discretely observed continuous-time non-linear systems”. Bayesian Analysis, 2008.
>
> [2] Deng et al.: “Continuous Latent Process Flows”. Neurips, 2021.
>
> [3] A.Bain and D.Crisan: “Fundamentals of stochastic filtering”. Springer, 2009.
>
> [4] Fearnhead et al.: “Random-weight particle filtering of continuous time processes”. Journal of the Royal Statistical Society, 2010.
>
> [5] Li, Xuechen, et al.: "Scalable gradients for stochastic differential equations." International Conference on Artificial Intelligence and Statistics. PMLR, 2020.
>
> [6] Tzen et al. : "Neural stochastic differential equations: Deep latent gaussian models in the diffusion limit." arXiv 2019.

---

> ### Author Response · Authors · 2024-06-13
> **Updates on Response to Reviewer xVMb**
>
> We would like to thank the reviewer again for the feedback. We've updated Figure 3 with a plot comparing the effective sample sizes of the latent samples under the scenarios with and without particle filtering in the revised version.

---

> > ### Comment · Reviewer_xVMb · 2024-06-14
> >
> > Hi, could you clarify what with/with of PF means? Does it mean with/without resampling?

---

> > > ### Author Response · Authors · 2024-06-14
> > > **Clarifications**
> > >
> > > Yes, it means with/without resampling.

---

### Author Response · Authors · 2024-06-11
**General Responses to all Reviewers**

**[Changes to the Paper]**

We would like to thank all reviewers for their helpful comments and constructive feedback. Based on their remarks and suggestions we have made the following changes to our manuscript (highlighted in blue in the revised paper):

- Introduction: we have clarified the problem setting and reworked the approaches taken in existing works (second paragraph). We have also included a brief and intuitive description of particle filtering (third paragraph).

- Preliminaries: to improve accessibility, we have added a subsection (2.1) that provides a generic and intuitive introduction to the particle filtering algorithm and describes the update step in particle filtering based on a discrete-time system.

- Overview figure: to improve accessibility, we have included an overview figure at the top of page 2 that describes the architectural setup, inference tasks, and high-level approach.

- Approach: we have made changes to some of the notation in the approach section to improve consistency. When describing the variational posterior process, we changed the notation to reflect a *parametrization* by the observations instead of a *conditioning* on the observations to avoid confusion with the true posterior process, which is conditioned on the observations. These changes are not explicitly highlighted in blue.

- Related work: we have added discussions on three existing continuous-time particle filtering works [1, 2, 3] and describe similarities and differences to our proposed framework.

- Experiments: additional experiments proposed by the reviewers (error bars, non-neural SDE with known dynamics, ESS plot) are currently running and will be added in a follow-up.

**[References]**

[1] A.Bain and D.Crisan: “Fundamentals of stochastic filtering”. Springer, 2009.

[2] T.Sottinen and S.Särkkä: “Application of Girsanov theorem to particle filtering of discretely observed continuous-time non-linear systems”. Bayesian Analysis, 2008.

[3] Fearnhead et al.: “Random-weight particle filtering of continuous time processes”. Journal of the Royal Statistical Society, 2010.

---

> ### Author Response · Authors · 2024-06-13
> **Updates on Additional Experiments for Reviewers**
>
> In the revised version of the paper, we updated Table 1 with mean and standard deviation valuations across multiple runs and Table 2 with 25th and 75th percentiles. In addition, we also added a plot comparing the effective sample sizes of the latent samples under the scenarios with and without particle filtering to Figure 3.

---

### Decision · Action_Editor_ZSi4 · 2024-07-12

**Recommendation:** Reject

**Comment:**

Given that there are doubts remaining about the claims of the paper, I think a further round of reviewing is needed, so I propose that the paper should not be accepted at this stage, but also recommend to the authors to revise and resubmit. In particular, the new revision should frame clearly in the introduction and in the main sections what is new to this paper and what is known work, rather than relegating this discussion to a few mentions in the related work section at the end of the paper, and potentially consider whether the experimental evaluation needs to be strengthened further.

**Audience:**

I believe that researchers working on latent-SDE models will find the paper interesting, as it brings methodology from the particle filtering literature to their attention. I'm less inclined to believe that the paper would be of interest to particle-filtering practitioners, given that much of the proposed framework is considered known by the reviewers.

**Claims And Evidence:**

My reading is that the paper makes two main claims:
1. that it introduces a framework for particle-filtering latent SDEs,
2. that the particle-filter framework outperforms commonly used alternatives.

Reviewer xVMb disagrees with claim 1, saying that the introduced framework has been explored extensively in literature, including textbooks. Reviewer L43e also raised concerns, saying that the paper is not clear at delineating what is new in section 3 and what is previous work. In response, the authors have updated section 5 (related work) with additional references, but not section 3. Considering the reviews and the authors' responses, I think the paper is still unclear as to which parts of section 3 are known results and what is new to this paper specifically. I believe a more thorough rewriting of section 3 is needed to address this, and for that reason I cannot consider claim 1 as being fully supported.

Regarding claim 2, both Reviewer xVMb and Reviewer L43e raised concerns about the level of rigour in the experimental evaluation. The authors responded by supplementing the empirical results with error bars and effective sample size plots. The reviewers have not repeated their concerns in their final recommendations, so I'm inclined to believe that they are satisfied with the updated version, although I would feel more confident if this were to be verified by a further round of reviewing.

**Resubmission Of Major Revision:**

The authors may consider submitting a major revision at a later time.